# AirPhyNet: Harnessing Physics-Guided Neural Networks for Air Quality Prediction

**Kethmi Hirushini Hettige**
School of Computer Science and Engineering
Nanyang Technological University, Singapore
kethmihi001@e.ntu.edu.sg

**Jiahao Ji**
School of Computer Science and Engineering
Beihang University, China
jiahaoji@buaa.edu.cn

**Shili Xiang**
Institute for Infocomm Research
A*STAR, Singapore
sxiang@i2r.a-star.edu.sg

**Cheng Long & Gao Cong**
School of Computer Science and Engineering
Nanyang Technological University, Singapore
{c.long,gaocong}@ntu.edu.sg

**Jingyuan Wang**
School of Computer Science and Engineering
Beihang University, China
jywang@buaa.edu.cn

## Abstract

Air quality prediction and modelling plays a pivotal role in public health and environment management, for individuals and authorities to make informed decisions. Although traditional data-driven models have shown promise in this domain, their long-term prediction accuracy can be limited, especially in scenarios with sparse or incomplete data and they often rely on *black-box* deep learning structures that lack solid physical foundation leading to reduced transparency and interpretability in predictions. To address these limitations, this paper presents a novel approach named Physics guided Neural Network for Air Quality Prediction (AirPhyNet). Specifically, we leverage two well-established physics principles of air particle movement (diffusion and advection) by representing them as differential equation networks. Then, we utilize a graph structure to integrate physics knowledge into a neural network architecture and exploit latent representations to capture spatio-temporal relationships within the air quality data. Experiments on two real-world benchmark datasets demonstrate that AirPhyNet outperforms state-of-the-art models for different testing scenarios including different lead time (24h, 48h, 72h), sparse data and sudden change prediction, achieving reduction in prediction errors up to 10%. Moreover, a case study further validates that our model captures underlying physical processes of particle movement and generates accurate predictions with real physical meaning. The code is available at: https://github.com/kethmih/AirPhyNet

## 1 Introduction

Air pollution refers to the disruption of natural traits of the atmosphere through contamination of the environment by chemical, physical, or biological substances. World Health Organization (WHO) reveals that nearly the entire global population (99%) inhales polluted air surpassing the WHO guideline limits (Kan et al., 2023). The resultant challenges have urged the need for advanced solutions in mitigating air quality issues. Thus, air quality prediction, involving the forecasting of future pollutant concentrations based on historical observations and other associated factors, has become a focal point of research.

In the literature, the air quality prediction problem is addressed under two primary paradigms: physics-based and data-driven. Physics-based models are inspired by atmospheric science and lever-

age domain-specific knowledge to formulate governing equations that can represent atmospheric processes. For example, Chemical Transport Models (CTMs) simulate atmospheric dynamics such as diffusion, advection, and deposition using ordinary or partial differential equations (Li et al., 2023; Jacob, 1999). However, these models involve solving complex differential equations and are often employed on larger spatial scales due to their higher computational overhead (Bauer et al., 2015; 2020). This restricts the real-time and fine-grained predictions. Moreover, the majority of these models require parameter calibration which may limit their ability to intricate real-world conditions such as complex human behaviors.

On the other hand, data-driven approaches utilize historical pollution data to learn complex relationships without the need for knowledge of the underlying physical processes. Among these methods, deep learning-based approaches have gained substantial attention over the past few years due to their enhanced expressive ability. For example, Spatio Temporal Graph Neural Networks (STGNNs) (Ji et al., 2023) which integrate GNNs and sequential models such as Recurrent Neural Networks (RNNs) have been employed to learn both spatial and temporal dependencies, thereby enhancing the prediction accuracy (Chen et al., 2021). Furthermore, recently attention-based models (Wang et al., 2022; Liang et al., 2023) have emerged as a robust alternative in the domain of air quality prediction specifically enhancing forecasts in the long term. However, despite the promising outcomes, these models also have their limitations. The majority of these models require extensive training data to achieve accurate long-term predictions. Furthermore, the absence of physics constraints in these models can limit their generalizability, potentially compromising their effectiveness in sparse data scenarios. Moreover, the inherent "black-box" nature of deep learning models raises challenges on interpretability which affects the decision-making in urban planning.

To alleviate the respective limitations of each paradigm, we present a novel hybrid learning approach: Physics guided Neural Network for Air Quality Prediction (AirPhyNet). This model attempts to leverage the strengths of both physics-based and data-driven methods contributing to a comprehensive framework, which can generate accurate air quality forecasts with a physical meaning. Specifically, the major motivation behind AirPhyNet is to use underlying physics related to air particle movement as domain knowledge to design an efficient and interpretable learning system. Accordingly, analogous to the differential equation-based traffic forecasting networks (Ji et al., 2022), in this work we exploit diffusion and advection processes within an end-to-end deep learning framework.

The overall AirPhyNet architecture consists of three components: $i$) an RNN-based encoder that extracts temporal information from the historical air quality data, $ii$) a GNN-based differential equation network that captures the air pollutant dispersion and flow dynamics, and $iii$) a decoder that generates the final pollutant concentrations based on the learned dynamics of physical processes. We evaluate our approach against several state-of-the-art baselines for forecasting air quality (PM2.5 concentrations in particular) up to 3 days ahead. Evaluations on two real-world benchmark air quality datasets show that AirPhyNet consistently outperforms baselines by a large margin, especially in sparse data scenarios. In summary, our contributions are as follows:

- We investigate the fundamental physics processes governing air pollutant transport and incorporate them into a comprehensive physics-guided deep learning framework. To the best of our knowledge, this is the first physics-guided hybrid deep learning approach of its kind introduced for outdoor air quality prediction.

- We propose a novel physics-guided deep learning model which integrates the physics-based dispersion processes and neural networks into one framework with a tailored GNN-based differential equation network.

- We conduct extensive experiments on two real-world air quality datasets and show that the proposed model outperforms the state-of-the-art models by a significant margin. Furthermore, a case study validates our model's ability to unveil the physical mechanism of air pollutant transport, thereby verifying its interpretability.

## 2 RELATED WORK

**Air Quality Prediction** Air quality prediction is one of the fundamental objectives within the scope of smart city and urban planning initiatives. Significant research efforts have been dedicated to

this domain over the years. Existing approaches can be primarily divided into two classes. (1) Physics based Models and (2) Data driven models. The former category generates predictions by emulating the dispersion of different air pollutants, relying on the principles of physical laws (Vardoulakis et al., 2003; Liu et al., 2005). These models portray the dynamics of air pollution, including processes like diffusion, advection, and chemical transformation as a set of differential equations, which are then solved through numerical simulations (Arystanbekova, 2004; Daly & Zannetti, 2007). However, they often fail to capture the real-world air pollution dynamics and demands for higher computational power. Recently, data-driven approaches have gained prominence, where historical observations and other contextual data (POI distributions, weather data) are used to capture spatiotemporal correlations (Han et al., 2021; 2023).The shallow machine learning methods (Suárez Sánchez et al., 2013; Su et al., 2023) have a limited capacity to model complex relationships while several deep learning models (Luo et al., 2019; Wang et al., 2022; Liang et al., 2023) have showed enhanced performance capturing complex spatiotemporal dependencies. However, these models require ample data for precise air quality prediction and they fail to encompass physical knowledge leading to limited generalization capability and interpretability.

**Physics Guided Deep Learning** Recently, several studies have been introduced to incorporate physical knowledge for data driven approaches (Raissi et al., 2017; Peng et al., 2022). While some of these approaches incorporate physical constraints into the loss function, other approaches present hybrid frameworks. For example, Jia et al. (2019) introduces a physics based regularized loss function in predicting lake temperature.Wang et al. (2020a) proposes a composite framework for turbulent flow prediction exploiting critical physical characteristics of the process while Ji et al. (2020) and Wang et al. (2023) present an interpretable model for grid based traffic flow prediction based on potential energy field. Similarly, Ji et al. (2022) presents a physics guided neural network approach for road network based traffic flow prediction. Moreover, Mohammadshirazi et al. (2023) introduces a framework which combines physics based state-space models with machine learning techniques for indoor air quality prediction. To the best of our knowledge, we introduce the pioneering work applying physics-guided deep learning in the realm of outdoor air quality prediction by proposing a hybrid learning framework, that integrates the physics-based dispersion processes into a deep learning framework with custom designed deep neural networks.

## 3 METHODOLOGY

### 3.1 AIR QUALITY PREDICTION PROBLEM

The goal of air quality prediction is to forecast future PM2.5 concentration based on historical concentrations and other related variables gathered from $N$ monitoring stations within a sensor network. We represent the sensor network as a directed graph $\mathcal{G} = (\mathbb{V}, E, W)$ where $\mathbb{V}$ is a set of nodes $|\mathbb{V}| = N$, $E$ is a set of edges and $W \in \mathbb{R}^{N \times N}$ is a weighted adjacency matrix indicating the proximity of nodes in terms of distance or flow field (see details in Section 2.3). A node $v_i \in \mathbb{V}$ represents an air quality monitoring station while $E_{i,j} \in E$ corresponds to a directed edge from node $v_i$ to node $v_j$ which is a potential pathway for air pollutant transport. Let $X_t \in \mathbb{R}^{N \times 1}$ and $P_t \in \mathbb{R}^{N \times 2}$ denote the node features at time step $t$. The air quality prediction problem aims to learn a function $h(.)$ that predicts concentration over the next $\tau$ steps minimizing prediction error:

$$\hat{X}_{(t+1:t+\tau)} = h(X_{(t-T+1:t)}, P_{(t-T+1:t)}, \mathcal{G}) \tag{1}$$

where $X_{(t-T+1:t)} \in \mathbb{R}^{T \times N \times 1}$ represents the historical PM2.5 concentrations, $P_{(t-T+1:t)} \in \mathbb{R}^{T \times N \times 2}$ represents the wind related attributes (i.e. wind speed and direction), $\mathcal{G}$ represents the corresponding graph network and $\hat{X}_{(t+1:t+\tau)} \in \mathbb{R}^{\tau \times N \times 1}$ represents the predicted future PM2.5 concentrations.

### 3.2 PHYSICAL DYNAMICS OF AIR POLLUTANT TRANSPORT

This work aims to design a physics guided deep learning model that incorporates physics based principles of air particle movement into a deep neural network. Thus, we first provide an overview of the physical techniques which are utilized on this idea and formulate them in the context of air pollutant transport as the foundation for this approach. All these techniques are derived based

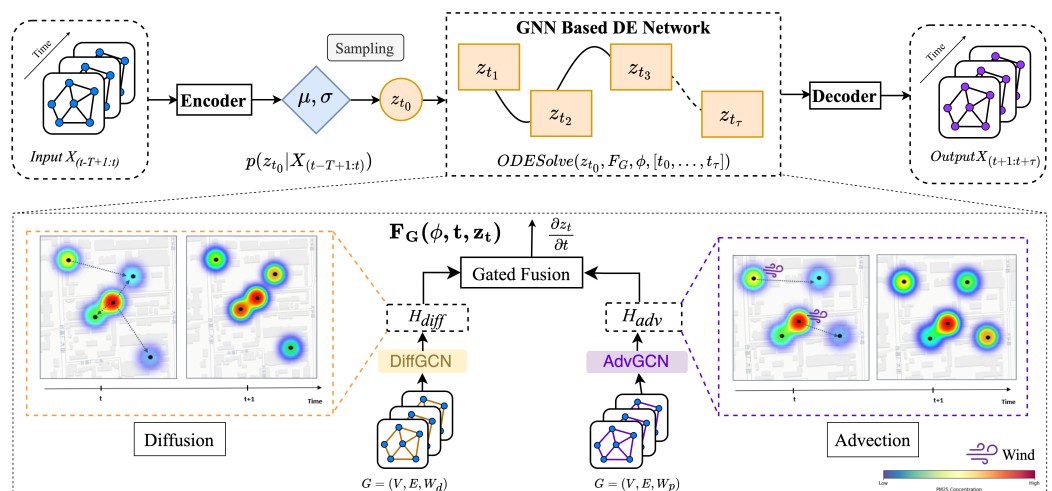

Figure 1: The overall architecture of AirPhyNet framework consisting of a RNN Encoder, GNN based DE Network and a Decoder. Heatmap indicate the PM2.5 concentrations while the dashed arrows represent the air pollutant transport between nodes due to diffusion and advection.

on the continuity equation which is one of the fundamental principles in physics and engineering. It describes the principle of conservation of mass within a closed system in mathematical form (Lienhard & Lienhard, 2008). In relation to our problem statement, here we use the continuity equation to describe the movement of particles through space as:

$$\frac{\partial X}{\partial t} + div\vec{F} = 0 \tag{2}$$

where $X$ is PM2.5 concentration, $\vec{F}$ is the flux of particles which describes the transport of PM2.5 particles and $div$ is the divergence operator. In simple terms, $div\vec{F}$ describes how the concentration changes at a particular point in space due to the transport of particles into or out of that point.

**Diffusion** is a key physical phenomenon that characterizes the random motion of particles from a high concentration place to a low concentration place within a medium. Diffusion process can be formulated using the continuity equation and Fick's Law (Grady et al., 2010). Eq. 2 explains the relation between the change of concentration of a substance and transport of that substance in space. On the other hand, Fick's Law (Paul et al., 2014) describes the flux of particles due to diffusion and it states that the magnitude of flux is directly proportional to the concentration gradient, i.e. $\vec{F} = -k\nabla X$ where $k$ represents the diffusion coefficient and $\nabla X$ represents the concentration gradient. In simple terms this equation indicates that the flux moves from high concentration regions to low concentration regions, with a magnitude proportional to the concentration gradient. By substituting the expression for the flux $\vec{F}$ into Eq.2 we obtain the diffusion equation for PM2.5 Concentration as follows:

$$\frac{\partial X}{\partial t} = k \, div\nabla X \tag{3}$$

**Advection** is another fundamental physical process that describes the transport of particles due to a flow field like bulk movement of air, typically driven by wind patterns or other large-scale air motions. In contrast to the diffusion process which explains the transport of particles due to the difference in concentration gradients, advection explains the transport of particles due to the effect of an external flow field. Thus, in an advection process the flux is formulated in terms of a vector field. i.e. $\vec{F} = \vec{v} X$ (Grady et al., 2010). By substituting this formulation of flux $\vec{F}$ into Eq.2 we obtain the the advection equation for PM2.5 Concentration as follows:

$$\frac{\partial X}{\partial t} = -div\,(\vec{v} X) \tag{4}$$

### 3.3 Diffusion-Advection Differential Equation

Air pollutant transport is a classical scenario which is governed by the aforementioned processes of diffusion and advection. Accordingly, in this section, we define these two processes on a sensor network and thereby propose an equation for change of PM2.5 concentration over space and time incorporating the air pollutant transport dynamics within a sensor network.

**Diffusion on a sensor network** refers to the dispersion of air pollutants within the network based on the concentration gradients between neighbouring sensor nodes. According to Bronstein et al. (2017), the graph laplacian operator $\Delta$ can be expressed in terms of the divergence of the gradient as $\Delta = -div\nabla$. Hence we re-write the diffusion equation (Eq.3) in terms of the graph laplacian as follows:

$$(\frac{\partial X}{\partial t})_{diff} = -k\,\Delta X = -\,k\,L\,X \tag{5}$$

where $L$ is the scaled laplacian computed using the distance based weighted adjacency matrix $W_d$ such that $W_d^{ij} = 1/d_{ij}$ where $d_{ij}$ is the haversine distance between nodes $i$ and $j$. In this study we consider the diffusion coefficient as $k = 0.1$ (Cussler, 2009).

**Advection on a sensor network** refers to the movement of air pollutants within a network, due to the effect of a flow field. To define advection on the sensor network, we follow the formulation proposed by Chapman & Chapman (2015). Accordingly, the discrete analogue of advection equation (Eq. 4) can be stated as:

$$\frac{\partial X^i}{\partial t} = \sum_{\forall j|j\to i} X^j v_{j\to i} - \sum_{\forall k|i\to k} X^i v_{i\to k} = -[M_{\mathcal{G}}X]^i \tag{6}$$

which states that the rate of change of PM2.5 concentration of node $i$ is the difference between the flow of particles in and out of the node due to the effect of the flow field. Here we consider $M_G$ as a modified laplacian determined using the flow field information as edge weights of the graph $\mathcal{G}$.

Generally, the flow field represents a vector field that characterizes the velocity of air particles at different locations and times within the system. In this work, we derive the flow field primarily based on wind speed and direction. Accordingly, we initially take the wind features $P_{(t-T+1:t)}$ and transform to a high dimensional space using a Multi Layer Perceptron (MLP). Then, we obtain the edge weight by the difference of the transformed wind representation of each source and destination node pair such that $W_p^{ij} = p^i - p^j$ where $p^i$ and $p^j$ are transformed wind representations of source node ($i$) and destination node ($j$) respectively (see Appendix A.2 for more details). Accordingly, the final advection differential equation can be stated in terms of scaled laplacian ($M$) computed using the flow-field based weighted adjacency matrix ($W_p$) as:

$$(\frac{\partial X}{\partial t})_{adv} = -[MX] \tag{7}$$

**Diffusion-Advection Differential Equation Function** In a real world scenario, the chemical concentration undergoes changes due to both of the above explained physical processes. To consider both diffusion and advection simultaneously, we use a gated fusion mechanism. This adaptive fusion process results in the following function which we name as the Diffusion-Advection Differential Equation (DE):

$$(\frac{\partial X}{\partial t}) = \alpha \odot (\frac{\partial X}{\partial t})_{diff} + (1-\alpha) \odot (\frac{\partial X}{\partial t})_{adv}$$

$$(\frac{\partial X}{\partial t}) = \alpha \odot H_{diff} + (1-\alpha) \odot H_{adv} \tag{8}$$

with

$$\alpha = \sigma(H_{diff}W_{\alpha,1} + H_{adv}W_{\alpha,2} + b_\alpha)$$

where $H_{diff}$ and $H_{adv}$ are the outputs of the diffusion and advection processes respectively, $W_{\alpha,1}$, $W_{\alpha,2}$ and $b_\alpha$ are learnable parameters, $\odot$ represents the element-wise product, $\sigma(.)$ denotes the sigmoid activation and $\alpha$ is the gate. Eq.8 describes the dynamic evolution of PM2.5 concentration within the sensor network.

### 3.4 Physics Guided Differential Equation Network

Based on the physical processes elaborated above and the derived Diffusion-Advection DE, we propose the Physics guided Neural Network for Air Quality Prediction (AirPhyNet). The model architecture of AirPhyNet (see Figure 1) is inspired by the seq2seq Neural ODE (Chen et al., 2018) which extends conventional neural networks into a continuous structure with ordinary differential equations. AirPhyNet consists of three main components: an RNN-based encoder to encode PM2.5 concentrations into an initial state, a GNN-based differential equation network governed by Eq. 8 that captures the physical dynamics of air pollutant transport and a decoder that generates the final PM2.5 concentrations based on the learnt dynamics of physical processes.

We initially employ a Gated Recurrent Unit (GRU) (Chung et al., 2014) as the RNN Encoder to extract the temporal information from the historical PM2.5 concentration sequence. Then, a fully connected network $g()$ is employed to determine the mean and standard deviation of $z_{t_0}$ from the final hidden states of GRU:

$$\{\mu_{z_{t_0}}, \sigma_{z_{t_0}}\} = g\left(GRU\left(X_{(t-T+1:t)}\right)\right) \tag{9}$$

Following (Ji et al., 2022), we determine the $z_{t_0}$ based on a reparameterization trick (Kingma & Welling, 2013), i.e. $z_{t_0} = \epsilon_i \mu_{z_{t_0}} + \sigma_{z_{t_0}}$ where $\epsilon_i$ is sampled from a standard normal distribution. Inspired by the neural ODEs, we then model the Diffusion-Advection DE in Eq. 8 as follows:

$$\frac{\partial z_t}{\partial t} = F_{\mathcal{G}}\left(\phi, t, z_t\right) \tag{10}$$

where $\phi$ represents trainable parameters including the diffusion coefficient $(k)$ and $F_{\mathcal{G}}$ is a neural network informed by the physics based DE in Eq. 8. Accordingly, the function $F_{\mathcal{G}}$ is formulated in terms of residual graph convolution network (GCN) layers in the form of :

$$F_{\mathcal{G}}\left(\phi, t, z_t\right) = -\alpha \odot k \odot Tanh\left(LX\right) - (1-\alpha) \odot Tanh\left(MX\right) \tag{11}$$

where graph $L$ and $M$ are the graph laplacian operators which use adjacency matrices with distance-based edge weights $(W_d)$ and flow field-based edge weights $(W_p)$ respectively. These operators calculate how the concentration of a particular node changes due to the diffusion and advection of pollutants from its neighbouring nodes. $Tanh()$ is the activation function (see Appendix A.3 for details).The results of the GCN layers corresponding to each process are aggregated using the gated fusion mechanism employed in Eq.8. Given the initial state $z_{t_0}$ and $F_{\mathcal{G}}$, we apply the neural ODE solver (Chen et al., 2018) to compute the future states $z_{(t_1:t_\tau)}$:

$$z_{(t_1:t_\tau)} = ODESolve\left(z_{t_0}, F_{\mathcal{G}}, \phi, [t_0, ..., t_\tau]\right) \tag{12}$$

Finally, the decoder module uses the output from the GNN-based DE Network representing the latent space, to generate predictions.It employs reshaping and aggregation steps to produce the desired output format with the correct dimensionality at each time step. AirPhyNet is trained through back-propagation by minimizing the Mean Absolute Error (MAE) (see A.4) between predicted values and ground truth values.

## 4 Experiments

### 4.1 Data Description

We assess the performance of our model on real world air quality datasets collected from two urban centres in China, Beijing and Shenzhen. Beijing dataset[1] has 35 major monitoring stations and data spans from 2017/01/01 to 2018/05/30 while the Shenzen dataset[2] has 11 monitoring stations and data spans from 2014/05/01 to 2015/04/30. Both datasets consist of hourly air quality and weather related observations which include concentrations of major pollutants (PM2.5, PM10, O3, NO2, SO2, and CO), temperature, pressure, humidity, wind speed and wind direction. Following majority of the previous work in the domain, this study focuses on PM2.5 concentration as the target variable. Additionally, we consider wind speed and wind direction as auxiliary variables. The missing values are imputed with the preceding 24-hour mean value of the respective station. We split each dataset chronologically in the ratio of 7:1:2 to generate distinct training, validation, and test sets respectively. Similar to the prior studies (Wang et al., 2020b; Liang et al., 2023) we consider 3 hours as a time step and use past 72-hour (24 steps) data to predict the future 72 hours.

---

[1]https://dataverse.harvard.edu/dataverse/whw195009
[2]https://www.microsoft.com/en-us/research/project/urban-air/

## 4.2 Experimental Settings

**Baselines** We compare our proposed AirPhyNet model with ten baselines under three categories. 1) Classical Methods: Historical Average (HA) and Vector Auto-Regression (VAR) 2) Spatio-temporal Deep Learning Methods:Diffusion Convolution Recurrent Neural Network (DCRNN) (Li et al., 2017), Spatio-temporal Graph Convolutional Network (STGCN) (Yu et al., 2017), GMAN (Zheng et al., 2020) and GTS (Shang et al., 2021) along with two powerful air quality prediction models PM2.5-GNN (Wang et al., 2020b) and AirFormer (Liang et al., 2023). 3) Differential equation network based methods: Latent-ODE (Rubanova et al., 2019) and ODE-LSTM (Lechner & Hasani, 2020). See Appendix A.5 for more details on baselines.

**Implementation Details** Our model is implemented in PyTorch 2.0.1 using NVIDIA GeForce RTX 3070 GPU. Adam optimizer is used for training the model. We set the batch size to 32 and use an initial learning rate of 5e-4 which decays over specific steps with a decay rate of 0.1. We use a GRU in the RNN encoder to encode the sequence and obtain the initial state. A grid search is conducted over $\{16, 32, 64, 128\}$, and 64 is chosen as number of hidden units of the GRU. In the ODESolver, we use *dopri5* as the numeral integration method with relative tolerance (rtol) and absolute tolerance (atol) set to 1e-5. Experiments on all deep learning models are conducted five times and the average results are reported. We use Mean Absolute Error (MAE) and Root Mean Square Error (RMSE) to evaluate the chosen baselines.

## 4.3 Performance Comparison

Table 1 shows the performance comparison of AirPhyNet with baselines over three time horizons on two datasets. Accordingly, AirPhyNet outperforms all competing baselines in both metrics, across all time horizons for both Beijing and Shenzen data. Compared to the second-best method, on average it achieves reduction in MAE and RMSE by 3.7% and 6.1% respectively which shows the effectiveness of employing physics based knowledge in a deep learning framework in capturing complex spatiotemporal relationships.

Table 1: Overall prediction performance comparison on metrics MAE and RMSE. The bold and underlined font show the best and the second best result respectively.

| Model | Beijing Data | | | | | | Shenzen Data | | | | | |
| | 24h | | 48h | | 72h | | 24h | | 48h | | 72h | |
| | MAE | RMSE | MAE | RMSE | MAE | RMSE | MAE | RMSE | MAE | RMSE | MAE | RMSE |
|---|---|---|---|---|---|---|---|---|---|---|---|---|
| HA | 38.37 | 83.91 | 45.8 | 95.56 | 50.58 | 101.51 | 8.35 | 12.52 | 9.72 | 14.34 | 10.54 | 15.39 |
| VAR | 60.10 | 102.92 | 60.44 | 103.02 | 60.64 | 103.07 | 21.50 | 26.09 | 21.84 | 26.50 | 22.17 | 26.92 |
| DCRNN | 35.99 | 52.55 | 49.66 | 67.50 | 57.01 | 74.67 | 7.76 | 11.54 | 9.99 | 14.11 | 10.98 | 15.14 |
| STGCN | 33.70 | 49.16 | 38.93 | 54.98 | 43.93 | 56.57 | 7.23 | 10.61 | 9.35 | 12.99 | 9.97 | 13.59 |
| GMAN | 50.62 | 66.05 | 50.73 | 66.07 | 50.69 | 65.87 | 9.76 | 12.70 | 10.08 | 13.22 | 10.07 | 12.99 |
| GTS | 34.99 | 51.45 | 54.18 | 71.87 | 73.50 | 89.59 | 6.58 | 9.55 | 8.70 | 12.15 | 10.54 | 13.94 |
| PM25GNN | 50.94 | 65.87 | 48.81 | 65.64 | 51.51 | 66.55 | 9.90 | 12.31 | 10.02 | 12.63 | 10.30 | 12.85 |
| AirFormer | 29.62 | 46.49 | 38.43 | 56.52 | 43.39 | 58.68 | 7.24 | 10.83 | 9.66 | 13.47 | 10.21 | 13.98 |
| LatentODE | 44.83 | 53.96 | 45.95 | 55.44 | 47.14 | 57.39 | 9.85 | 12.30 | 10.24 | 12.89 | 10.73 | 13.33 |
| ODE-LSTM | 46.19 | 57.56 | 49.18 | 62.39 | 51.45 | 63.66 | 10.55 | 13.24 | 11.36 | 14.18 | 12.03 | 14.81 |
| AirPhyNet | **29.11** | **42.16** | **36.69** | **48.66** | **42.23** | **53.07** | **6.38** | **9.48** | **8.18** | **11.38** | **9.51** | **12.51** |
| % Improvement | 1.73 | 9.33 | 4.53 | 11.49 | 2.66 | 6.19 | 3.01 | 0.76 | 5.97 | 6.38 | 4.58 | 2.64 |

From Table 1 we can also observe that: 1) Deep-learning based approaches outperform the classical methods including HA and VAR demonstrating their capability of learning complex relationships. 2) Spatio-temporal Deep Learning approaches initially designed for traffic forecasting such as GTS and STGCN demonstrate strong adaptability for the air quality forecasting problem and yield competing results as the state of the art air quality prediction methods, such as the AirFormer.3) Neural DE network based approaches have demonstrated less effective performance compared to spatio-temporal deep learning methods due to their inability to capture spatial relationships which are vital for accurate air quality prediction. All in all, these results verify the ability of AirPhyNet in achieving precise air quality predictions while leveraging the physics based domain knowledge of air particle movement.

We also evaluated the performance of models in predicting sudden changes in air quality. Compared to the baselines, AirPhyNet shows reduction prediction errors upto 8% in sudden change prediction exhibiting its' effectiveness in understanding the system in terms of physical processes and thereby predicting abrupt shifts in air quality.Complementary to the primary evaluations, we additionally conducted experiments to assess how well our model generalizes to unseen data, which is identified as a common advantage of physics-based models. Based on the results, AirPhyNet has outperformed all baselines across all time horizons by a significant margin. Specifically, the MAE and RMSE have been reduced by around 8% on average compared to the second best method. This emphasizes the superiority of our model's ability to generalize effectively and to achieve precise long term air quality predictions particularly in scenarios with limited data (i.e. regions with limited number of monitoring stations). See A.6 for detailed results.

## 4.4 ABLATION STUDY

**Effect of Physical knowledge:** To further illustrate the effect of incorporating physical knowledge we compared AirPhyNet with two other variants. a)*AdvOnly* model which only considers the advection process b)*DiffOnly* model which only considers the diffusion process . Fig. 2a shows the results comparison for Beijing dataset.

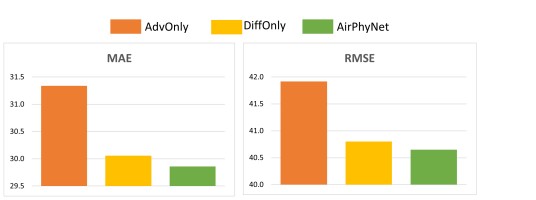
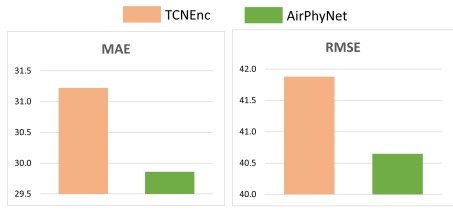

(a) Effect of Physical Knowledge in AirPhyNet          (b) Effect of Encoder

Figure 2: Results of Ablation Study

According to the results, we observe that AirPhyNet outperforms both *AdvOnly* and *DiffOnly* models, demonstrating the advantage of leveraging physical knowledge to predict air quality. Moreover, *DiffOnly* model performs better than *AdvOnly* model which shows that the advection process alone cannot adequately capture the the air pollutant transport dynamics while the diffusion process has shown better capability in doing so. However, *DiffOnly* model exhibits slightly lower performance compared to AirPhyNet due to the absence of the advection component. Thus, it highlights the importance of identifying the appropriate combination physical processes to comprehensively address the problem.

**Effect of Encoder:** To study the efficacy of the RNN-based encoder in capturing temporal dependencies, we compare our model with another variant *TCNEnc* which replaces the RNN encoder with a Temporal Convolutional Network (TCN) based encoder. According to the results in Fig.2b, we observe that AirPhyNet outperforms TCNEnc model by relatively by a large margin, demonstrating the advantage of utilizing GRU in capturing temporal dependencies.

## 4.5 CASE STUDY

In this section, we conduct a case study to demonstrate the strong performance of our approach qualitatively in terms of the two physical processes diffusion and advection. To illustrate the interpretability with regard to advection, we visualize the predicted PM2.5 concentrations together with the wind direction attributes for both datasets.Fig. 3 show the spatial distribution of monitoring stations of the Beijing dataset and Shenzen dataset at two timesteps which are four hours apart. In Beijing dataset, wind flows towards the southwest direction in the respective times. It can be observed that the PM2.5 particles have moved towards the direction of the wind with time, specifically in the circled regions. PM2.5 concentration has reduced in the circled region over the four-hour period and the particles have moved and joined the PM2.5 cloud heading towards the southwest wind direction. Similarly in Shenzen dataset, the wind flows in the East direction. PM2.5 concentration

has increased over the four-time steps in the circled region and joined the PM2.5 cloud moving towards the east wind direction.

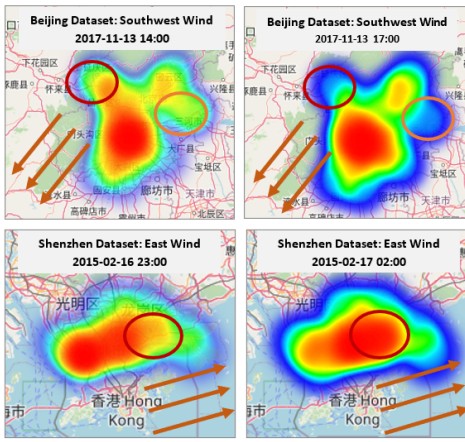

Figure 3: Visualization of predicted PM2.5 concentrations and wind direction. Heatmap represents the PM2.5 concentrations and the arrows indicate the wind direction

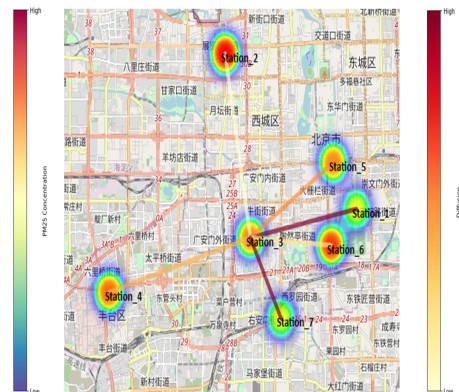

Figure 4: Visualization of predicted PM2.5 concentrations and diffusion from Station_3 on 2017-11-13 14:00.The line segments represent diffusion from Station_3 to its neighbouring stations and the magnitude of diffusion is reflected by the line colour.

Next, we visualize the predicted PM2.5 concentrations and magnitude of diffusion.Fig. 4 shows the spatial distribution of a subset of stations of the Beijing dataset where the PM2.5 concentrations predicted by the AirPhyNet are represented by a heat map and diffusion from Station_3 to its neighbouring stations is represented by line segments. The visualization shows that the diffusion of particles from Station_3 is relatively higher to stations with low concentrations (Station_1, Station_7) while it's low to stations with higher concentration (Station_2). Therefore, the predicted concentrations governed by physical process capture real mechanism of physical diffusion process which states that the particles move from places with high concentration to low concentration. Thus, the predicted concentrations can be interpreted in terms of diffusion. In summary, the findings in this section affirm that AirPhyNet successfully captures the underlying physical principles of particle movement and yields precise predictions with a physical meaning.

## 5 CONCLUSION AND FUTURE WORK

In this paper, we introduced an approach for air quality prediction based on the application of diffusion and advection based differential equations which serve as the foundational principles governing the air pollutant transport in the atmosphere. We propose a novel Physics guided Neural Network known as AirPhyNet that leverages these fundamental physical equations and seamlessly incorporates them into a deep learning architecture. Experiments conducted on two real world datasets show that AirPhyNet reduced prediction errors by a significant margin (4%-10%) compared to the existing methods. A case study further affirms model's capability to generate precise predictions that can be interpreted within a physical context. However, the applicability of our approach to highly chemical-reacting pollutants remains uncertain as such complex chemical behaviours are not captured by the model. In the future, we plan to expand the applicability of our model across a wide spectrum of air pollutants by further augmenting it with chemical kinetics. It could enhance the predictive capabilities for pollutants governed by complex chemical processes. Moreover, we aim to explore its capabilities for cross-city air quality prediction through transfer learning, PM2.5 source tracking and plume modelling.

ACKNOWLEDGEMENT

This work is partially supported by the National Research Foundation, Prime Minister's Office, Singapore under the Aviation Transformation Programme. Kethmi Hirushini Hettige's work is supported by the Singapore International Graduate Award. Jingyuan Wang's work is supported by the National Natural Science Foundation of China No. 72222022. We would also like to express our sincere gratitude to the anonymous reviewers for their invaluable feedback, which greatly contributed to the improvement of this manuscript.

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

## A  APPENDIX

### A.1  NOTATION

### A.2  COMPUTATION OF LAPLACIAN MATRICES FOR DIFFUSION AND ADVECTION

On the sensor network $\mathcal{G} = (\mathbb{V}, E, W)$ with $N$ monitoring stations we define two weighted adjacency matrices. The weighted adjacency matrix based on distance ($W_d$) determines the edge weights as follows:

$$W_d^{ij} = 1/d_{ij}$$

where $d_{ij}$ is the haversine distance between nodes $i$ and $j$. On the other hand, the weighted adjacency matrix based on flow field ($W_p$) determines the edge weights as follows:

$$p^i = FlowNet(P^i)$$

$$p^j = FlowNet(P^j)$$

$$W_p^{ij} = p^i - p^j$$

Table 2: Summary of Notations used in the paper

| Symbol | Description |
|---|---|
| $\mathcal{G}$ | Graph |
| $N$ | Number of Nodes in the Sensor Network |
| $\mathbb{V}, \mathrm{v}_i$ | Nodes of the Graph and $i$-th node |
| $E$ | Edges of the Graph |
| $W_d, W_d^{ij}$ | Distance based weighted adjacency matrix and its entries |
| $W_p, W_p^{ij}$ | Flow-field based weighted adjacency matrix and its entries |
| $X_t$ | PM2.5 Concentration at timestep $t$ ($X_t \in \mathbb{R}^{N \times 1}$) |
| $P_t$ | Wind features at timestep $t$ ($P_t \in \mathbb{R}^{N \times 2}$) |
| $\vec{F}$ | Flux of Particles |
| $k$ | Diffusion Coefficient |
| $\vec{v}$ | Vector Field |
| $L$ | Scaled laplacian computed based on $W_d$ |
| $M$ | Scaled laplacian computed based on $W_p$ |

where $P^i$ and $P^j$ represent the wind related attributes of source node ($i$) and destination node ($j$) and *FlowNet* is a MLP used to obtain the flow field representation from corresponding wind data.

Eq.5 is formulated to define the physical process of diffusion within a sensor network. In other words, it describes the change in concentration due to the dispersion of air pollutants within the network. Thus the distance based scaled graph laplacian $L$ is computed as follows:

$$\overline{L} = I - D^{-\frac{1}{2}} W_d D^{-\frac{1}{2}}$$

$$L = 2\overline{L}/\lambda_{max} - I$$

where $\overline{L}$ is the laplacian matrix, $W_d$ is the distance based weighted adjacency matrix, $D$ is the diagonal degree matrix and $\lambda_{max}$ is the largest eigen value of $\overline{L}$.

Similarly, Eq.7 is formulated to define the physical process of advection within a sensor network which describes the change in concentration due to the movement of air pollutants within the network due to the effect of a flow field. Accordingly, the flow field based scaled graph laplacian $M$ is computed as follows:

$$\overline{M} = I - D^{-\frac{1}{2}} W_p D^{-\frac{1}{2}}$$

$$M = 2\overline{M}/\lambda_{max} - I$$

where $\overline{M}$ is the laplacian matrix, $W_p$ is the distance based weighted adjacency matrix, $D$ is the diagonal degree matrix and $\lambda_{max}$ is the largest eigen value of $\overline{M}$.

### A.3 GCNs used in formulation of $F_{\mathcal{G}}$

As stated in Eq. 11, $F_{\mathcal{G}}$ is a neural network guided by the Diffusion-Advection DE. $F_{\mathcal{G}}$ is formulated in terms of residual graph convolution network (GCN) layers as follows.

**GCN for Diffusion Process**

$$H_{diff}^{(l)} = \sigma \left( \sum_{k=0}^{K-1} \theta_k \cdot L^k \cdot H_{diff}^{(l-1)} \right)$$

$$H_{diff} = \sum_{l=0}^{L'} H_{diff}^{(l)}$$

where $H_{diff}^{(l)}$ is the updated state at layer $l$, $\sigma$ is the activation function, $\theta_k$ are the Chebyshev coefficients, and $L^k$ represents the $k$th power of the scaled laplacian calculated using distance based

weighted adjacency matrix $W_d$ . $H_{diff}$ represents the final output of the GCN designed to capture the diffusion process.

**GCN for Advection Process**

$$H_{adv}^{(l)} = \sigma \left( \sum_{k=0}^{K-1} \theta_k \cdot M^k \cdot H_{adv}^{(l-1)} \right)$$

$$H_{adv} = \sum_{l=0}^{L'} H_{adv}^{(l)}$$

where $H_{adv}^{(l)}$ is the updated state at layer $l$, $\sigma$ is the activation function, $\theta_k$ are the Chebyshev coefficients, and $M^k$ represents the $k$th power of the scaled laplacian calculated using flow field based weighted adjacency matrix $W_p$ . $H_{adv}$ represents the final output of the GCN designed to capture the advection process.

### A.4 EVALUATION METRICS

Let $x = (x_1, \ldots, x_n)$ represents the ground truth, and $\hat{x} = (\hat{x}_1, \ldots, \hat{x}_n)$ represents the predicted PM2.5 concentration values. The evaluation metrics we used in this paper are defined as follows:

**Mean Absolute Error (MAE)**

$$\text{MAE}(x, \hat{x}) = \frac{1}{n} \sum_{i=1}^{n} |x_i - \hat{x}_i|$$

**Root Mean Square Error (RMSE)**

$$\text{RMSE}(x, \hat{x}) = \sqrt{\frac{1}{n} \sum_{i=1}^{n} (x_i - \hat{x}_i)^2} \tag{13}$$

### A.5 DETAILED SETTINGS OF AIRPHYNET AND BASELINES

**AirPhyNet:** In our model we use a GRU as the RNN encoder to encode the sequence and obtain the initial state. We conduct a grid search over $\{16, 32, 64, 128\}$, and choose 64 as the number of hidden units of the GRU. A fully connected layer with 50 hidden units is then used to transform the hidden states of GRU and obtain the mean and the standard deviation of the initial state. In the ODESolver, we use *dopri5* as the numeral integration method with relative tolerance (rtol) and absolute tolerance (atol) set to 1e-5. Adam optimizer is used for training the model. We set the batch size to 32 and use an initial learning rate of 5e-4 which decays over specific steps with a decay rate of 0.1. We also employ an early stop strategy with a patience of 20, allowing for a maximum of 100 epochs.

**HA:** Historical Average is a tradtional statistical method which predicts the PM2.5 concentration by computing the average value of historical readings in corresponding time intervals.Here, we use 1 day as the period and consider the average of the historical concentration of the same time interval from previous 4 days as prediction.

**VAR:** VAR model is implemented using the *statsmodels* python packages and the number of lags is set to 3.

**DCRNN** Diffusion Convolution Recurrent Neural Network introduced in the domain of traffic forecasting uses bidirectional random walks on the graph structure to capture the spatial dependencies while an encoder-decoder architecture is used to capture the temporal dependencies. We use two recurrent layers each with 64 hidden units, in both encoder and decoder. Initial learning rate is set to 0.01 and it is reduced with a rate of decay rate of 0.1 starting at the 20th epoch. The maximum steps of random walks $(K)$ is set to 3.

**STGCN** Spatio-temporal Graph Convolution Network integrates graph convolutions with gated convolutions to capture spatio-temporal relationships as opposed to employing regular convolutional

and recurrent units. It consists of two ST-Conv blocks and the channels the three layers of each block are 64, 16, 64 respectively. The kernel size of graph convolution ($K$) and temporal convolution ($K_t$) is 3. The initial learning rate is set at 1e-3 and it decays with a rate of 0.7 after every 5 epochs.

**GMAN** Graph Multi-Attention network is introduced for the task of traffic prediction which consists of an encoder-decoder architecture with multiple spatio-temporal attention blocks. An adam optimizer with an initial learning rate of 1e-3 is used to train the model. The number of STAttention blocks ($L$) is set to 3 and use 8 attention heads (($K$)) each having a dimensionality of 8 (($d$)).

**GTS** Graph for time series is designed to simultaneously learns a graph structure among multiple time series to enhance time series forecasting. Adam is used as the optimizer and initial learning rate is set to 1e-3. Dropout rate is considered as 0.2, the value of k in kNN is set to 30 and the weight regularization is 1.

**PM2.5-GNN** PM2.5-GNN is one of the powerful domain knowledge enhanced networks introduced to predict PM2.5 concentration. Although the original model employs multiple features to integrate domain knowledge, we only consider wind speed and wind direction features in this study. RMSprop is used as the optimizer with a learning rate of 5e-4.

**AirFormer** Airformer is a transformer based air quality prediction model designed to forecast air quality across thousands of locations simultaneously. Initial learning rate is set to 5e-4 and it decays with a rate of 0.5 after every 3 epochs. Adam optimizer is used to train the model. For the dartboard spatial attention module, we partition the space by two circles with radius 50km and 200km respectively. We use 4 Airformer blocks in the model and set the hidden dimension to 32.

**Latent-ODE** This model extends the capabilities of RNNs by introducing continuous-time hidden dynamics governed by ordinary differential equations (ODEs). It uses ODE-RNN as the encoder. The initial learning rate is set to 5e-4 and the latent dimension of the GRU is 100. *dopri5* is used as the numeral integration method with relative tolerance (rtol) and absolute tolerance (atol) set to 1e-5.

**ODE-LSTM** ODE-LSTM is introduced to address the limitations of recurrent neural networks (RNNs) with continuous-time hidden states in modeling irregularly-sampled time series by introducing a novel algorithm that separates memory from continuous-time dynamics. Here we set the latent dimension to 64, Explicit Euler is used as the ODE solver iswith relative tolerance ((rtol) and absolute tolerance (atol) set to 1e-5. The learning rate is 5e-4.

All the baseline models were trained both with and without wind-related attributes (i.e., wind speed and wind direction) as auxiliary variables. However, with the exception of PM2.5-GNN, the rest of the baseline models exhibited degraded performance when wind-related attributes were incorporated in the respective models. Therefore, the reported baseline performances do not include wind-related attributes, except for PM2.5-GNN.

### A.6    ADDITIONAL EXPERIMENTAL RESULTS

**Sudden Change Prediction:** In alignment with the studies by  Zheng et al. (2015) and  Liang et al. (2023), we also evaluated the performance of deep learning based methods in predicting sudden changes of air quality. Accordingly, the sudden changes are defined as instances where PM2.5 concentration exceeds 50 μg/m³ and 20 μg/m³ and exhibits a change of more than ±20 μg/m³ in the following three hours for Beijing Data and Shenzen Data respectively. As illustrated in Table 3 AirPhyNet shows the best performance in predicting sudden changes in both datasets. AirFormer (Liang et al., 2023) has also shown competing performance compared to the other baselines due to its' utilization of stochastic latent spaces. Compared to the second best method (i.e. AirFormer), AirPhyNet shows an average reduction of 6.2% and 4.6% in MAE and RMSE respectively.

**Sparse Data Prediction:** In order to assess the generalizability of our approach, we evaluated the models using a smaller training dataset with train:val:test split as 3:1:6 and obtain the results shown in Table 4. Based on the results, AirPhyNet has outperformed all baselines across all time horizons by a significant margin. Specifically, the MAE and RMSE have been reduced by 8.3% and 8.1% on average respectively compared to the second best method.

Table 3: Sudden Change prediction performance comparison on metrics MAE and RMSE. The bold and underlined font show the best and the second best result respectively.

| Model | Beijing Data | | Shenzhen Data | |
|---|---|---|---|---|
| | MAE | RMSE | MAE | RMSE |
| DCRNN | 18.616 | 53.0629 | 3.3468 | 8.0377 |
| STGCN | 21.821 | 56.8061 | 3.2087 | 8.1773 |
| GMAN | 48.2294 | 80.4201 | 4.6336 | 9.9652 |
| GTS | 22.5072 | 63.1401 | 3.7127 | 8.1299 |
| PM25GNN | 28.196 | 69.6186 | 4.9186 | 10.2459 |
| AirFormer | 14.1553 | 48.4329 | 3.1317 | 7.9183 |
| LatentODE | 43.5 | 76.8365 | 3.549 | 8.639 |
| ODE-LSTM | 38.248 | 76.2235 | 3.6311 | 8.6356 |
| AirPhyNet | **13.4987** | **47.7057** | **2.8904** | **7.3148** |
| % Improvement | 4.64 | 1.50 | 7.71 | 7.62 |

Table 4: Sparse Data prediction performance comparison on metrics MAE and RMSE. The bold and underlined font show the best and the second best result respectively.

| Model | Beijing Data | | | | | | Shenzen Data | | | | | |
|---|---|---|---|---|---|---|---|---|---|---|---|---|
| | 24h | | 48h | | 72h | | 24h | | 48h | | 72h | |
| | MAE | RMSE | MAE | RMSE | MAE | RMSE | MAE | RMSE | MAE | RMSE | MAE | RMSE |
| DCRNN | 44.24 | 56.03 | 55.80 | 68.13 | 64.90 | 78.24 | 7.99 | 11.48 | 10.11 | 13.99 | 11.06 | 15.01 |
| STGCN | 33.73 | 44.90 | 39.93 | 50.51 | 46.62 | 57.53 | 7.25 | 10.50 | 8.96 | 12.13 | 9.75 | 12.92 |
| GMAN | 46.61 | 55.59 | 47.60 | 56.62 | 48.65 | 57.63 | 9.69 | 12.35 | 9.88 | 12.68 | 9.91 | 12.64 |
| GTS | 38.09 | 48.97 | 60.28 | 71.45 | 80.42 | 92.06 | 6.60 | 9.61 | 8.70 | 11.87 | 10.09 | 13.37 |
| PM25GNN | 47.34 | 61.55 | 46.33 | 60.81 | 47.64 | 61.65 | 10.78 | 13.18 | 10.84 | 13.22 | 10.90 | 13.23 |
| AirFormer | 31.47 | 43.20 | 41.64 | 53.22 | 44.50 | 56.39 | 7.14 | 10.22 | 8.93 | 12.17 | 9.25 | 12.51 |
| LatentODE | 44.02 | 51.40 | 45.94 | 53.28 | 47.68 | 54.51 | 9.81 | 12.15 | 10.01 | 12.38 | 10.25 | 12.54 |
| ODE-LSTM | 49.95 | 61.16 | 51.93 | 63.19 | 53.44 | 64.19 | 10.36 | 13.26 | 11.01 | 13.97 | 11.49 | 14.39 |
| AirPhyNet | **27.99** | **38.48** | **35.78** | **45.82** | **40.45** | **50.37** | **6.47** | **9.50** | **7.75** | **10.76** | **8.63** | **11.58** |
| % Improvement | 11.08 | 10.93 | 10.39 | 9.29 | 9.09 | 10.68 | 1.88 | 1.09 | 10.92 | 9.34 | 6.73 | 7.43 |

## A.7 DISCUSSION

According to the results presented in this study, AirPhyNet reveals its consistent performance in air quality prediction over competing baselines across various metrics and time horizons. Notably, the model excels in capturing complex spatiotemporal relationships, showcasing a significant reduction in MAE and RMSE compared to alternative methods. The incorporation of physics-based knowledge into a deep learning framework proves advantageous, particularly in predicting sudden changes in air quality. Additionally, the model exhibits robust generalization to unseen data, highlighting its potential for application in regions with limited monitoring station data.Furthermore, the comparison with variants focusing solely on advection or diffusion processes underscores the model's ability to comprehensively address the air pollutant transport dynamics while the case study further validates this fact. All in all, the AirPhyNet model has shown significant capability in yielding precise air quality forecasts with a physical meaning.

These findings hold significant implications for air quality prediction. AirPhyNet's performance with lower prediction errors, suggests its potential real-world applicability in scenarios where precise forecasts are crucial. The ability to generalize well to unseen data implies its' adaptability across diverse environments, making it a useful tool for regions with limited monitoring infrastructure. Moreover, the integration of physics-based knowledge not only enhances prediction accuracy but also provides insights which could be beneficial in decision making related to urban planning.

While the results are promising, it is vital to acknowledge certain limitations in this study. Our focus solely on PM2.5 concentration, while significant, may not account for the complete spectrum of pollutants that contribute to overall air pollution. There exist pollutants with diverse chemical properties and reactivity patterns which might exhibit behaviors that are not captured by our model. Therefore,

the applicability of our method to highly chemical-reacting pollutants remains uncertain. Additionally, our deep learning architecture emphasizes the incorporation of physical processes, limiting its suitability for pollutants heavily influenced by chemical reactions.This could potentially hinder the model's accuracy in scenarios dominated by chemical reactions rather than physical transport.

To address these limitations, future studies could broaden the scope to predict multiple air pollutants and determine the overall air quality. Integration of chemical reaction information into the modeling framework, possibly by incorporating chemical kinetics with deep learning, could enhance the predictive capabilities for pollutants governed by complex chemical processes. These models can be further enhanced to capture the interplay between different domains through insightful collaboration with experts in chemistry and environmental science. Another potential future direction is to leverage this architecture for cross-city air quality prediction. This involves training our model on data from one city and adapting it to predict air quality in a different city through appropriate fine-tuning and transfer learning techniques.

