# OpenReview forum: "AirPhyNet: Harnessing Physics-Guided Neural Networks for Air Quality Prediction"
_ICLR.cc/2024/Conference — ICLR 2024 poster_

### Official Review · Reviewer_tKfZ · 2023-10-15

**Soundness:** 3 good
**Presentation:** 3 good
**Contribution:** 3 good
**Rating:** 6
**Confidence:** 4

**Summary:**

This paper considers the task of air quality prediction (PM2.5) simultaneously over a network of weather monitoring stations, using wind direction and speed as covariates. This is achieved with the help of an enoder-decoder architecture which can represent past values of the target variables and covariates, including the spatial correlations between the different stations. The encoded structure is used to solve ODEs based on advection and diffusion through a suitably designed Graph Neural Network, which can represent the spatial relations between the different stations. Instead of using computationally expensive Chemical Transport Models (CTM), this physics-based NN model can predict the state at the end of the prediction period (24/48/72 hours) and this is used to generate the final predictions of PM2.5 using decoder. It is shown that the proposed model can perform better than existing NN-based models and traditional statistical models for air quality prediction, including in situations where there is abrupt air quality changes. Some intuition is also provided through a case study, where the pollution hot-spots are found to shift in the direction of winds.

**Strengths:**

The paper proposes an architecture that achieves two things: i) solves differential equations for advection and diffusion over a network using a GNN with Neural ODE solver, and ii) uses this to make spatio-temporal predictions of air quality using wind direction+speed as covariates.

The strengths of the paper lies in the facts that:
i) the proposed framework is quite novel
ii) the reported results are very strong

**Weaknesses:**

The weak points of the work are mostly in the experimental part:

i) The station sizes are relatively small (35 and 11). It is not clear how the method will scale to bigger networks
ii) The comparisons provided are mostly against other neural network based methods. The work is motivated by stating that CTMs are very expensive. Yet, no comparison with CTM in term of either computational cost or accuracy is provided. Two "traditional methods" are mentioned (HA and VAR), but no details of those are provided.
iii) There is no analysis of the nature of the data that is being dealt with.

Apart from these, the analysis leaves a few loose ends, mentioned in the next section.

**Questions:**

1) What was the architecture of the "decoder"? I did not find this anywhere.
2) What is the significance of the encoded state "z" ? Is it possible to visualize it? Is it possible to run the Neural ODE on the original data itself, without the encoding?
3) What is the impact of the lookback window size T?
4) The model seems to have been trained separately for the two cities in question. Is it possible to train it on one city and use it on the other? Which parameters will need fine-tuning in that case?
5) Is the data periodic in nature? How strong is the spatial correlation between the different stations?
6) Two covariates are considered- wind direction and wind speed. Do the competitor methods also use these? How will the proposed model perform if these covariates are not considered? Or if more covariates are considered?
7) How are the two "traditional" methods - HA and VAR implemented? Specifically, are they done specific to each station separately? If so, it may not be a fair comparison.
8) How does the proposed approach compare with CTM?

---

> ### Author Response · Authors · 2023-11-18
> **Addressing Experimental Concerns, Comparison with CTM and clarification of effect of model components.**
>
> Thank you for the insightful review. We appreciate the acknowledgment of our novel architecture and strong results. In response to your feedback, the noted concerns are addressed below.
>
> ### Addressing Weaknesses:
> 1. **Model Scalability:** We acknowledge your concern about the relatively smaller number of stations considered in our experiments. The primary reason for conducting experiments in a limited number of stations is the unavailability of comprehensive public datasets at the monitoring station level.  We explored multiple available datasets (such as the [KnowAir Dataset](https://https://github.com/shuowang-ai/PM2.5-GNN)) where data is available at a coarser granularity. However, given our emphasis on fine-grained predictions at the monitoring station level, we opted the current datasets which have also been used in majority of the similar work below.
>     * Du, S., Li, T., Yang, Y., & Horng, S. J. (2019). **Deep air quality forecasting using hybrid deep learning framework**. IEEE Transactions on Knowledge and Data Engineering, 33(6), 2412-2424.
>     * Wang, C., Zhu, Y., Zang, T., Liu, H., & Yu, J. (2021, March). **Modeling inter-station relationships with attentive temporal graph convolutional network for air quality prediction**. In Proceedings of the 14th ACM international conference on web search and data mining (pp. 616-634).
>
>  We also conducted some experiments on a sample dataset provided by Airformer Model upto 150 nodes. We vary the number of nodes and measure the training time and inference time. The results shown [here](https://anonymous.4open.science/r/RebuttalDiagrams-23DC/Scalability.png) exhibit the training and inference time of our model as the number of nodes increase. However, we assumed airformer model's adjacency matrix weights include distance information since the explicit location details are not available for this dataset. We hope that the generated trend line could be insightful to reflect the scalability of our model despite of the limitations and assumption.
> We acknowledge the importance of extending our experiments to a broader geographical scope and intend to explore collaborations and avenues to acquire additional datasets for future research.
>
> Weaknesses 2 and 3 are addressed below.
>
> ### Addressing Questions:
>
> 1.	**Decoder:**  We acknowledge the concern with regard to the decoder architecture. Accordingly, we clarify the decoder module under the last paragraph of Section 3.4 in the revised manuscript as follows.
>
>     *Finally, the decoder module uses the output from the GNN-based DE Network representing the latent space, to generate predictions. It employs reshaping and aggregation steps to produce the desired output format with the correct dimensionality at each time step. AirPhyNet is trained through backpropagation by minimizing the Mean Absolute Error (MAE) (see A.4) between predicted values and ground truth values.*
>
> 2.	**Significance of the encoded state $z$ :** The significance of this encoded state lies in its ability to capture and represent the  temporal dependencies in the input data. Direct visualization of z might be challenging as it is in a higher dimensional space. However, it can be transformed into a lower dimension for visualization and used to gain insights on the learned temporal representations.
>
>     It is possible to apply Neural ODEs directly to the original data without an intermediate encoding step. However it might result in a more complex and computationally demanding model, as the network has to learn to capture critical temporal dynamics without an encoder.
>
> 3.	**Impact of the Lookback window:** As shown in the [LookbackWindow visualization](https://anonymous.4open.science/r/RebuttalDiagrams-23DC/LookbackWindow.png), the error reduces as the lookback window increases, since a wider temporal range is captured to make the predictions. As stated in Section 4.1, similar to the prior studies (Wang et al., 2020b; Liang et al., 2023) we use past 72-hours as the lookback window.
>
> **Please find the clarifications for the remaining questions in the next response window due to the character limitation per response.**

---

> ### Author Response · Authors · 2023-11-18
> **Addressing Experimental Concerns, Comparison with CTM and clarification of effect of model components.**
>
> 4. **Cross city Knowledge Transfer:**  We appreciate your insightful question, which could be a valuable extension to our model. Training our model in one city and applying it to another city which can be referred to as cross-city generalization. It is possible to implement taking into account certain considerations such as:
>     * **Spatial Differences:** Different cities have different spatial characteristics, such as road layouts, building distributions, and overall urban structures.
>     * **Different Sensor Networks:** Sensor networks in different cities may use different types of sensors, have varying sensor densities.
>
>     With these considerations we can learn a similarity function among the sensor networks of each city considering spatial features. Then the model trained on a source city can be finetuned based on this similarity function and data from the target city. Since the physical processes are invariant across cities, no specific adjustments are needed inside the existing model architecture. Similar approaches have been proposed in the traffic prediction domain: [Wang et al.,2018](https://arxiv.org/pdf/1802.00386). We included this idea as a plausible future work in our Conclusion and Future work section of the revised manusctipt. Thank you for pointing this out.
>
> 5.	**Exploratory Analysis of Data:** Both datasets used in this study are periodic in nature and has around 6.5% and 1% missing values in Beijing and Shenzhen data, respectively. These data also include sudden changes in air quality concentrations. Strong spatial correlations exist in both the datasets. Corresponding timeseries plots and correlation plots can be found [here](https://anonymous.4open.science/r/RebuttalDiagrams-23DC/).
>
> 6.	**Effect of Wind Data:** As stated under Appendix 5 in our manuscript, all the baseline models were trained both with and without wind-related attributes. However, with the exception of PM2.5-GNN, the rest of the baseline models exhibited degraded performance when wind-related attributes were incorporated in the respective models. Therefore, the reported baseline performances do not include wind-related attributes, except for PM2.5-GNN.
>
>     The proposed model without wind data is similar to the DiffOnly Model which do not consider the advection component as illustrated in the Ablation study. DiffOnly model exhibits a slightly degraded performance to our overall model. The results are included in the Ablation study in Section 3.4.
>
>     If more covariates are considered, they can be used as auxiliary features in the model. Additional covariates were not considered in our experiments, since they are not essential to model the physical processes. We tried to achieve optimal performance with less data since most of the weather data are not often available at the monitoring station level. This can be considered one of the advantages of our model.
>
> 7.	**Implementation Details of HA and VAR:** Further implementation details of these models are included in Appendix (A.5) of the revised manuscript.
>
> 8.	**AirPhyNet Comparison with CTM:** We appreciate your observation regarding the absence of a direct comparison with Chemical Transport Models (CTMs). Widely used CTMs such as CMAQ (Community Multiscale Air Quality) and WRF-Chem (Weather Research and Forecasting with Chemistry) simulate wide range of atmospheric dynamics in addition to diffusion and advection which demands for multiple input variable including land use data, boundary conditions, meteorological parameters like boundary layer height. Furthermore, these models require sophisticated parameter calibrations and higher computational resources. Due to these concerns CTMs were not used in this study for analysis. Instead, we introduce AirPhyNet as a potential solution for the aforementioned challenges.

---

> > ### Comment · Reviewer_tKfZ · 2023-11-18
> >
> > I thank the authors for their detailed responses to my questions/comments. A few concerns, however, remain for me:
> >
> > 1) Use of auxiliary variables: I do not see any logical reason why most competing models should perform worse in presence of wind speed as additional input, than in its absence. The question arises if incorporation of the auxiliary inputs to these models were done in the best possible way. Otherwise, some insights about why these models are unable to utilize additional inputs would be useful to appreciate the results.
> >
> > 2) I am aware that running the CTM has many practical challenges, but are there no archived datasets of past simulations available? Without some sort of comparison with CTM, ML-based approaches may not find acceptance in the environmental science community which probably prefers to trust process-based models like the CTM more at this point (at least that has been my personal experience)
> >
> > 3) The authors mention that the data is periodic in nature. Periodic data is generally easier to forecast than non-periodic data. Can the proposed model work well even if the data were non-periodic?

---

> > > ### Author Response · Authors · 2023-11-20
> > >
> > > We appreciate your follow-up comments and concerns regarding our work. Please find further clarifications below.
> > >
> > > 1. **Use of Auxiliary Variables:** We acknowledge your concern about the performance of competing models in the presence of wind speed and wind direction as additional inputs.
> > >     * **Integration of Auxiliary Variables:** As stated in our previous response, all the baseline models we used for comparison were trained both with and without wind-related attributes. The integration of wind attributes is adhered to the original implementations of each model. More elaborately, in the PM2.5GNN model, the wind variable was incorporated into the graph structure as edge weights, whereas in the other baseline models, it was treated as an additional feature.
> > >
> > >       However, it is important to note that the baseline models, in their original implementations, incorporate several other meteorological attributes such as temperature, humidity, and pressure as auxiliary variables. We did not consider all these attributes, since our physics based model is not directly associated these variables. The distinctive performance of these baseline models may be attributed to the combined representative ability of these diverse meteorological variables. It is plausible that the inclusion of temperature, humidity, and pressure, alongside wind-related data, contributes to the enhancement of the models' predictive capabilities within their specific context.
> > >
> > >     * **Configuration of wind variables:** In both our model and the PM25GNN model, the wind variable is integrated into the graph structure as edge weights. This differs from the baselines, where wind is treated as an additional feature. This variation in the treatment of the wind variable within the graph structure may contribute to the observed differences in performance.
> > >
> > >
> > >      In summary, the AirPhyNet model leverages on physical processes that are not directly linked to other meteorological variables. Consequently, these variables were excluded from our analysis. The auxillary variables used (i.e Wind Attributes) were incorporated to our model in the best possible way to capture their influence on the underlying physical processes.
> > >
> > >     However, we acknowledge the importance of the further examining the effect of these multiple meteorological variables. Therefore, we aim to explore further on how these multiple variables can be integrated into our existing model through different physical processes or otherwise, as future work. We appreciate your attention to this aspect.
> > >
> > > **Please find the clarifications for the remaining questions in the next response window due to the character limitation per response.**

---

> > > ### Author Response · Authors · 2023-11-20
> > >
> > > 2. **Comparison with CTM:** We understand your concern about the lack of comparison with a Chemical Transport Model (CTM).
> > >     * **Archived Datasets and Spatial Granularity**: We searched for simulated datasets of PM2.5 concentrations for the specific location and time dimensions relevant to our study. Unfortunately, we could not locate datasets specifically tailored to our requirements. There are available datasets for PM2.5 concentrations in the contiguous US; however, it's worth noting that these datasets come with a coarser resolution, utilizing 12km grids ([Berrocal et al., 2020](https://www.sciencedirect.com/science/article/am/pii/S1352231019307691),[Lightstone et al, 2017](https://www.mdpi.com/2073-4433/8/9/161/pdf))
> > >
> > >         Considering the objective of our model to achieve precise fine-grained predictions at the monitoring station level, it's important to acknowledge that CTMs like CMAQ operate on a grid,and the resolution of this grid determines the spatial scale at which predictions are made. Despite the capability of CMAQ to run at relatively high resolutions, the finest scales may still be on the order of kilometers. This means that the model may not capture localized variations at the scale of individual monitoring stations. This is why CMAQ is valuable for understanding regional and large-scale air quality patterns as opposed to finegrained predicsions.
> > >
> > >         To address the finer scales inherent to individual monitoring stations, users frequently complement CMAQ with additional tools, such as machine learning or deep learning models ([Li et al., 2023](https://www.nature.com/articles/s41612-023-00475-3)).
> > >
> > >         If a CTM-based simulated dataset for our specified region and time period become available, we are open to considering a broader spatial grid (i.e. region) , for which the CTM data is available, as a set of virtual stations. This approach will involve calculating the features of virtual stations by summarizing the features of physical stations within the designated region. Each of these regions can then be treated as a node in a graph, allowing us to apply our proposed model and compare its predictions with CTM-generated values. If you are aware of any such datasets, we would greatly appreciate your recommendation, and we can try to conduct the experiment as outlined.
> > >
> > >         Furthermore, we would like to note that, there are numerous studies in literature ([Lightstone et al, 2017](https://www.mdpi.com/2073-4433/8/9/161/pdf),[Li et al., 2023](https://www.nature.com/articles/s41612-023-00475-3)) based on simulated data in different other regions at a coarser resolution. These studies have demonstrated the enhanced performance of Deep Learning or Hybrid Deep Learning approaches compared to CTMs.
> > >     * **Limitations of CMAQ/CTM Modeling:** As specified in my previous response, several studies highlight and prove through empirical evaluations the challenges CTMs face. For example, [Rao et al. (2019)](https://acp.copernicus.org/articles/20/1627/2020/) show the inability of CTMs in consistently reproducing stochastic variations in the atmosphere, a factor where data-driven approaches like ours exhibit notable strengths. Moreover, [Kitayama et al. (2019)](https://www.sciencedirect.com/science/article/pii/S1352231018307696), [Rao et al. (2019)](https://acp.copernicus.org/articles/20/1627/2020/) have shown the uncertainties arising from parameterization leading to significant biases, including overestimations or underestimations, in concentration predictions.
> > >
> > >
> > > In light of these considerations, our approach aims to address the limitations associated with both CTMs and traditional machine learning models. However for certain highly reactive air pollutants such as NO, CO2 etc, physics-guide deep learning itself will not be effective. As noted under the conclusion of our revised manuscript, one of the limitations of our work is its constrained applicability to highly chemical-reacting pollutants as such complex chemical behaviours are not captured by the model.
> > >
> > > In this paper, we focused on PM2.5 using a physics-guided deep learning approach, demonstrating enhanced predictive accuracy and resolution while maintaining interpretability. We believe innovative methodologies incorporating chemical reactions/ chemical kinerics are essential for addressing the additional challenges posed by highly reactive air pollutants.
> > >
> > > **Please find the clarifications for the remaining questions in the next response window due to the character limitation per response.**

---

> > > ### Author Response · Authors · 2023-11-20
> > >
> > > 3. **Model performance for Non- Periodic Data:** We acknowledge the inherent advantages that periodic data often provides in forecasting; however, we would like to emphasize that our proposed model is designed to leverage to capture complex spatiotemporal dependencies, even in scenarios where the data may exhibit non-periodic characteristics.
> > >
> > >     As stated in section 4.3 and illustrated in detail in Appendix 6, our model has demonstrated robust performance sudden change predictions. In such situations, there are abrupt shifts where the air quality patterns do not strictly adhere to periodic cycles. This showcases our model's ability to discern and adapt to irregularities, accommodating variations that may not follow a clear periodic trend.

---

> > > > ### Comment · Reviewer_tKfZ · 2023-11-22
> > > >
> > > > I thank the authors for their responses. My opinion about the paper is positive.

---

> ### Author Response · Authors · 2023-11-22
> **Request for Reconsideration of the score**
>
> Dear Reviewer tKfZ,
>
> We are happy that our responses have effectively addressed your concerns. We sincerely appreciate the time and effort you have dedicated to reviewing our work. Especially your suggestion extending our methodology for cross city transfer learning is indeed a promising avenue for future research. It helped us to  broaden our future research directions.
>
> As we approach the end of the author/reviewer discussions within the next few hours, we kindly request your reconsideration of the score and the associated confidence level in light of the responses and revisions we have provided.
>
> Thank you so much for devoting time to improving our work!

---

### Official Review · Reviewer_8KYF · 2023-10-31

**Soundness:** 4 excellent
**Presentation:** 3 good
**Contribution:** 4 excellent
**Rating:** 8
**Confidence:** 5

**Summary:**

This paper introduces AirPhyNet, a physics-informed graph neural network for air quality forecasting. Specifically, a diffusion-advection differential equation is first established to represent the physical process of air particle movement. Then, a physics-guided model is proposed to capture air pollution dynamics and generate physically consistent forecasting results by seamlessly integrating the predefined differential equation into a graph neural network. Experimental results on two real-world benchmark datasets demonstrate the superiority of AirPhyNet over several state-of-the-art baseline models in various forecasting scenarios. Moreover, a case study is also included to show the potential interpretability of the proposed model.

**Strengths:**

1. As far as I know, this is the first few attempts to combine physical modeling and deep learning for air quality forecasting.
2. It is reasonable to take the advection and diffusion of air pollutants into consideration when building forecasting models.
3. The paper is well-organized and easy to follow. There are also enough experiments to demonstrate the effectiveness and interpretability of the proposed method.

**Weaknesses:**

1. The difference between diffusion and advection is unclear. From my understanding, they all describe the transport of air pollutants over space and time. Although the authors briefly explained the difference in Section 2.2, the details are not clearly stated. More discussions would be better.
2. In Equation 9, the authors claim they adopt a reparametrization trick to derive the hidden representation z_{t_0}. The rationale behind this choice is not fully explained. Why not directly use the final hidden state of GRU as z_{t_0}?
3. In Section 2.4, the authors mentioned they leverage a decoder to generate the prediction results, but it lacks sufficient explanations. For example, the instantiation of the decoder should be illustrated in the paper. Moreover, the loss function should be formally defined.
4. Some existing forecasting methods, such as DCRNN and AirFormer used in this paper, can also model the diffusion and advection process to some extent. So, what’s the major advantage of injecting physical principles into machine learning models? More discussions are appreciated.
5. There are some typos. For example, in Section 3.1, “Shanghai” should be “Shenzhen”.

**Questions:**

See weakness

---

> ### Author Response · Authors · 2023-11-17
> **Clarification on Diffusion and Advection, Elaboration of Reparameterization Trick and Detailed advantages of integrating physical principles into Deep Learning Models.**
>
> Thank you for the insightful and constructive feedback. We appreciate your positive remarks on AirPhyNet's innovation and experimental demonstration.
>
> ### Addressing Weaknesses:
> 1.	**Clarification on Diffusion and Advection:** We understand your concern about the clarity of the distinction between diffusion and advection. As you’ve stated, both these processes describe the transport of air pollutants over space and time. The distinction lies in the transport mechanism. Diffusion explains transport of particles among locations due to the difference in concentration gradient while the advection explains the transport of particles due to the effect of an external flow field like wind. In the revised manuscript, we added a clearer distinction in section 2.2 as follows:
>
> *In contrast to the diffusion process which explains the transport of particles due to the difference in concentration gradients, advection explains the transport of particles due to the effect of an external flow field.*
>
> 2. **Reparameterization Trick:** 	We appreciate your insightful question regarding the adoption of the reparameterization trick. The main motivation to use this trick arises from the training and generative capabilities of Variational Autoencoder (VAE) framework.  The use of the reparameterization trick in our model architecture allows for the introduction of stochasticity into the encoding process. Directly using the final hidden state of GRU as ${z_{t_0}}$ , would result in in a deterministic representation, limiting model's ability to capture the inherent uncertainty in the latent space. By introducing the reparameterization trick, we enable the model to sample from a distribution defined by mean and standard deviation, providing a stochastic element to the latent representation.
>
>     In summary, this trick is useful in enhancing model’s robustness and flexibility as it enables to account for variability in the temporal patterns of PM2.5 concentration, acknowledging that a single deterministic hidden state might not fully capture the complexity and uncertainties.
>
> 3.	**Decoder and Loss Function:** Based on your concern, we clarify the decoder module and loss function under section 3.4 of the revised manuscript as follows. The formulation of the loss functions is included in the appendix due to the constraints of the page limit.
>
> *Finally, the decoder module uses the output from the GNN-based DE Network representing the latent space, to generate predictions.It employs reshaping and aggregation steps to produce the desired output format with the correct dimensionality at each time step. AirPhyNet is trained through backpropagation by minimizing the Mean Absolute Error (MAE) (see A.4) between predicted values and ground truth values*
>
> 4.	**Advantages of Integrating Physical Principles:** We acknowledge the importance of addressing the unique contributions of our approach in comparison to existing forecasting methods, such as DCRNN and Airformer. As you’ve stated principles like Diffusion is reflected in the DCRNN model even though it is not explicitly utilized as a physics principle, but Airformer model does not incorporate these processes as it uses an attention-based transformer architecture which does not utilize a Graph Neural Network structure. Furthermore, we would like to highlight the following major advantages of injecting physical principles into machine learning models, as opposed to data driven methods like DCRNN:
>     * **Enhanced Interpretability:** By incorporating a physics-informed framework, AirPhyNet provides enhanced interpretability in its predictions. The physical principles guide the learning process, allowing for a more transparent understanding of the underlying mechanisms influencing air quality dynamics.
>     * **Improved Generalizability:** Physics-informed models are designed to capture underlying physical processes, which can improve the generalization capability of the model to unforeseen conditions or locations. While data-driven models like DCRNN and AirFormer rely heavily on available training data, our approach leverages physics-based priors to enhance the model's ability to make accurate predictions in diverse scenarios.
>     * **Domain Knowledge Integration:** Injecting physical principles allows us to integrate domain knowledge into the learning process. This incorporation of expert knowledge ensures that the model adheres to the fundamental laws governing air quality dynamics.
>     * **Consistency with Physical Laws:** The physics-informed approach ensures that the generated forecasts are consistent with known physical laws governing the movement and dispersion of air pollutants. This consistency enhances the reliability of predictions, especially in situations where traditional machine learning models may struggle due to their lack of adherence to these fundamental principles.
>
> 5.	**Typos:** Thank you for pointing this out. We reviewed and fixed these typos in the revised manuscript.

---

> > ### Comment · Reviewer_8KYF · 2023-11-22
> > **Thanks for the response**
> >
> > Thanks for the clarification. I am positive to this paper.

---

> > > ### Author Response · Authors · 2023-11-22
> > > **Note of thanks**
> > >
> > > Dear Reviewer 8KYF,
> > >
> > > We are happy that our responses have effectively addressed your concerns. We would like to express our sincerest gratitude once again for taking the time to review our paper and provide us with such detailed and constructive feedback!

---

### Official Review · Reviewer_6p5Q · 2023-11-01

**Soundness:** 3 good
**Presentation:** 2 fair
**Contribution:** 2 fair
**Rating:** 6
**Confidence:** 3

**Summary:**

In the presented study, the authors address the limitations of traditional data-driven models for air quality prediction, which often lack long-term accuracy and transparency due to their black-box deep learning nature. They introduce a novel approach called AirPhyNet, which integrates well-known physics principles of air particle movement (diffusion and advection) into a neural network using differential equation networks and a graph structure. This method not only enhances the model's interpretability by tying it to real-world physics but also shows superior performance on real-world datasets, outperforming state-of-the-art models in various test scenarios and reducing prediction errors by up to 10%. The model's ability to accurately capture the underlying physical processes of particle movement is further validated through a case study.

**Strengths:**

1. Good writeup style, in terms of grammar and readability. In Particular, the methodology has a comparatively good readability.

**Weaknesses:**

1. Research question is not specified. It is better to specify it.
2. Before diving into the main work, the related work should be discussed.
3. Section 4: Air Quality Prediction: Most of the papers from this subsection that have been cited, are outdated. I would request to add critical discussions of all such works and how the present approach overcomes them.
For instance:
a. https://ieeexplore.ieee.org/document/10152272
b. https://arxiv.org/abs/2308.03200
c. https://www.nature.com/articles/s41598-022-12355-6
d. https://www.sciencedirect.com/science/article/pii/S1309104223000715?casa_token=1NXW1K1A37EAAAAA:NOxq1SvOhxDOOuqWmSssZAMZYUeApCukMcQGYNRWgAkeNKWBamlEBoWke0IfgmZNpPBtT3vElOc
e. https://ieeexplore.ieee.org/abstract/document/9877800
4. There is no discussion section. I would highly suggest adding it.
5. There is no limitation mentioned in the paper.

**Questions:**

1. Minor spacing issues in sentences. For instance, in abstract.
2. I did not see the full form of DE Network. If you are using any abbreviation, please make sure to introduce the full form of it along with the short form. For instance, “Differential Equation (DE) is something. DE does that……”.
3. I would suggest you shift the section of Related Work before Methodology, after introduction.
4. Some of the discussions from the findings are mentioned in Section 3.4 (in the first paragraph of Page 8). However, I would suggest you add it in the discussion section (as mentioned in Weakness #4) along with the possible reasons behind the claims.
5. Please add the limitation subsection.

---

> ### Author Response · Authors · 2023-11-16
> **Research Question Specification, Justification of Related work and Inclusion of Discussion and Limitations**
>
> Thank you for your thoughtful review. We appreciate your feedback and have addressed your suggestions and concerns below.
>
> ### Addressing Weaknesses
> 1. **Research Question specification:** We would like to draw your attention to Section 2.1 where we have specified our research question. This section and specifically Equation 1, elaborates on the air quality prediction problem with the corresponding notations.We also specify the rationale behind the air quality prediction problem in section 1, highlighting the limitations of existing techniques. We further discuss the potential of integration of physics dynamics into deep learning networks to enhance the interpretability and predictive accuracy of the problem.
>
> 2. **Related Work Discussion:** We would like to clarify that in our current manuscript structure, a brief overview of related work is presented in the Introduction section. We provide a comprehensive discussion of related work in a dedicated section later in the manuscript. To address your concern, we have moved the Related Work section after the Introduction and before the Methodology in our revised manuscript.
>
> 4. **Citations:** Regarding your concern on the citations, we accept the fact that some of the citations, specially under “Air Quality Prediction” subsection are outdated. That is because we have included the progressive development of techniques within this domain from initial physics-based methods to the current state-of-the art data driven methods. All these methods are within our technical scope.
>
>     We appreciate your suggestions on some up-to-date related work, and we have included two of them (Su et al., 2023, Wang et al., 2022) together with some new citations under related work of the revised manuscript. However, we would like to emphasize that the other suggested works are out of the scope of this work. The first two works (Nilesh et al, 2022; Mondal et al 2023), involve image processing methods for Air Quality Prediction which is distinct from our study which focuses on leveraging physics-guided deep learning for forecasting tasks, specifically tailored to monitoring station data. The last work (Lin et al, 2023) involves anomaly detection, which differs from our primary focus on air quality forecasting task.
>
> Weaknesses 4 and 5 are addressed below.
>
> ### Addressing Questions
> 1.	**Spacing Issues:** We carefully reviewed the manuscript and adjusted the minor spacing issues present. Thank you for pointing this out.
>
> 2.	**Abbreviation:** The full form of Differential Equation (DE) is introduced in section 2.3 under the Diffusion-Advection Differential Equation Function, when it is first mentioned in the manuscript to enhance clarity for readers.
>
> 3.	**Section Order:** We have moved the Related Work section after the Introduction and before the Methodology in our revised manuscript.
>
> 4.	**Discussion Section:**  We appreciate your suggestion on the discussion section, and it has been included in the revised manuscript in the Appendix (A.7) due to the constraints on the page limit. The discussion section includes summary of results, implications of the findings in general, limitations and few recommendations for further enhancement.
>
> 5.	**Limitations:** Limitations are included within the discussion section in the revised manuscript as given below. We also highlighted these limitations within section 5 (Conclusion and Future Work). A seperate subsection was not included due to the constraints on the page limit.
>
> *While the results are promising, it is vital to acknowledge certain limitations in this study. Our focus solely on PM2.5 concentration, while significant, may not account for the complete spectrum of pollutants that contribute to overall air pollution. There exist pollutants with diverse chemical properties and reactivity patterns which might exhibit behaviors that are not captured by our model. Therefore, the applicability of our method to highly chemical-reacting pollutants remains uncertain. Additionally, our deep learning architecture emphasizes the incorporation of physical processes, limiting its suitability for pollutants heavily influenced by chemical reactions.This could potentially hinder the model’s accuracy in scenarios dominated by chemical reactions rather than physical transport.*
>
> Please note that the revised manuscript is uploaded and all the revisions are indicated in red font color.

---

> > ### Comment · Reviewer_6p5Q · 2023-11-23
> > **Acknowledging and Thanks for Respons**
> >
> > I thank the authors for their detailed responses to my questions/comments. I updated my rating to a 6. I am positive about the paper.

---

> ### Author Response · Authors · 2023-11-22
> **Request for Reconsideration of the score**
>
> Dear Reviewer  6p5Q,
>
> We sincerely appreciate the time and effort you have dedicated to reviewing our work. We have addressed the your questions and concerns above. Given that the end of author/reviewer discussions is just in few hours, this is to kindly ask for your reconsideration of the score. We would really appreciate it if our next round of communication could leave time for us to resolve any of your remaining or new questions.
>
> Thank you so much for devoting time to improving our work!

---

> ### Author Response · Authors · 2023-11-23
> **Note of Thanks**
>
> Dear Reviewer 6p5Q,
>
> Thank you for your feedback and engagement with our paper. We're pleased to hear that our detailed responses to your questions and comments have contributed to a positive impression. Your updated rating of 6 is much appreciated, and we are grateful for your constructive input.
>
> Best Regards,
> Authors

---

### Official Review · Reviewer_pK4K · 2023-11-04

**Soundness:** 3 good
**Presentation:** 4 excellent
**Contribution:** 2 fair
**Rating:** 6
**Confidence:** 3

**Summary:**

The study introduces AirPhyNet, a physics-guided neural network designed for enhanced air quality prediction. This method incorporates fundamental physics principles into the network architecture, improving predictive performance and interpretability. For this, it draws from existing literature in physics guided ML and neural ODEs. Tests on real-world data showcase its potential to improve over existing methods.

**Strengths:**

- Putting together multiple complex concepts and methods is indeed a difficult task and requires a thorough understanding of physical dynamics and deep learning.

- The paper addresses a significant and timely problem.

- The narrative is clear and accessible.

- The case study illustrates some of the physics that the model captures.

**Weaknesses:**

- My main concern with this work is the lack of contributions to hybrid AI or AI in general. Authors did not identify any technical gaps in our current hybrid AI methods. Instead, authors take what other researchers have developed for physics-guided ML in a variety of domains (e.g., physics) and use them for air quality prediction. Therefore, the method appears to be a combination of multiple well known methods with some developments in how to incorporate the specific physics priors for air quality priors. The air quality priors are just new equations and do not pose a significant technical challenge. Therefore, I do not think this is not a significant contribution for ICLR's research track. Perhaps the paper's contribution is better suited to a domain journal or the applied track of an AI conference.

- Liang et al. 2023 (cited by authors in experimental setup) performed experiments in 342 cities in China and data appears publicly available. However, authors of this paper performed experiments only in 2 cities.

**Questions:**

- What is the reason for selecting so few cities?

---

> ### Author Response · Authors · 2023-11-15
> **Contribution of AirPhyNet to the field of hybrid AI, Justification of the choice of a limited number of cities and Significance of the work for ICLR's research track.**
>
> ### Addressing concerns on Contributions:
> We appreciate your concern about the perceived lack of contribution to hybrid AI or AI in general. Allow us to clarify the technical gaps and the specific contributions our work brings to the field to bridge these gaps. As highlighted in Section 1, Paragraph 2, the existing deep learning models including hybrid AI models impose technical challenges on interpretability, generalizability and require ample amount of data for precise predictions. Thus, as opposed to the existing hybrid data driven approaches, we introduce synthesis of physics dynamics and deep learning methods for air quality prediction which is a non-trivial task. Thereby some of our unique contributions are as follows:
>
> * **Integration of Physics Principles with Deep Learning to develop a comprehensive architecture:**
>     * The incorporation of physics priors within a deep learning architecture for air quality predictions, as presented in our paper, involves more than just introducing new equations or fusing exisitng architectures. It involves tailoring and leveraging existing deep learning methodologies to represent physics dynamics which we believe is a significant contribution.
>     * We draw inspiration from existing literature on physic-based (Section 3.2), **define these physics dynamics of Air Pollutant transport on a sensor network** (Section 3.3) and **seamlessly integrate them into a deep learning architecture** with custom designed adjacency matrices and neural network layers(Section 3.4).
>     * Even though, there exist methods which use these physical priors for air quality prediction (such as CTM), they have not  been explicitly combined with deep learning architectures. Our application and evaluation of these concepts in a real-world context represents a novel and challenging endeavour.
>
> * **Leveraging the model with Interpretability:** The interpretability of deep learning models in diverse urban problems such as air quality prediction is a significant technical challenge that has not been adequately addressed. Our model, AirPhyNet not only enhances the predictive performance but also provides forecasts which are interpretable based on physics dynamics, thus contributing to a better understanding of the model’s decision-making process.
>
> With the aforementioned contributions and technical challenges addressed, we believe our work aligns well with the goals and interests of the ICLR research track.
>
> ### Addressing concerns on Experimentation:
> The primary reason for conducting experiments in a limited number of cities is the unavailability of comprehensive public datasets. As you pointed out [Liang et al. (2023)](https://ojs.aaai.org/index.php/AAAI/article/view/26676/26448), whose work we cited in our experimental setup utilized their own dataset covering mainland China. Only 20 instances of each training, validation and test sets of this dataset are made accessible which would not yield comparable and meaningful results.
>
> We would also like to draw your attention to the [KnowAir Dataset](https://https://github.com/shuowang-ai/PM2.5-GNN) which is another publicly available dataset we explored but did not use in our analysis. This dataset provides air quality and meteorological data at the city level. Given our emphasis on fine-grained predictions at the monitoring station level, we opted the Beijing and Shenzhen Datasets which have also been used in majority of the past work in the literature*. However, we acknowledge the importance of extending our experiments to a broader geographical scope and intend to explore collaborations and avenues to acquire additional datasets for future research.
>
> *Given below are some of the latest works in literature which have used the Beijing and/or Shenzhen Datasets:
> * Du, S., Li, T., Yang, Y., & Horng, S. J. (2019). **Deep air quality forecasting using hybrid deep learning framework**. IEEE Transactions on Knowledge and Data Engineering, 33(6), 2412-2424.
>     * Conducted experiments on two datasets from Beijing covering 1 station and 36 stations respectively.
> * Han, J., Liu, H., Zhu, H., Xiong, H., & Dou, D. (2021, May). **Joint air quality and weather prediction based on multi-adversarial spatiotemporal networks**. In Proceedings of the AAAI Conference on Artificial Intelligence (Vol. 35, No. 5, pp. 4081-4089).
> * Wang, C., Zhu, Y., Zang, T., Liu, H., & Yu, J. (2021, March). **Modeling inter-station relationships with attentive temporal graph convolutional network for air quality prediction**. In Proceedings of the 14th ACM international conference on web search and data mining (pp. 616-634).
>     * Conducted experiments on two datasets from Beijing and Tianjin covering 35 stations and 26 stations respectively.
> * Han, J., Liu, H., Zhu, H., & Xiong, H. (2023). Kill Two Birds with One Stone: **A Multi-View Multi-Adversarial Learning Approach for Joint Air Quality and Weather Prediction**. IEEE Transactions on Knowledge and Data Engineering.

---

> > ### Comment · Reviewer_pK4K · 2023-11-21
> > **Thanks for response**
> >
> > Thank you for the clarifications. Definitely, building such framework for this important application is an excellent contribution. My claim is that this statement is not for a technical contribution to AI. I think there probably are technical contributions, but authors do not claim them explicitly.
> >
> > Authors say that "tailoring and leveraging existing deep learning methodologies to represent physics dynamics (...) is a significant contribution." Here the challenge is "define these physics dynamics of Air Pollutant transport on a sensor network" and then "integrate them into a deep learning architecture".
> >
> > These are too general. I could say the same about any other hybrid AI method for an application with spatiotemporal diffusion (say traffic). My question here is: what were the specific technical challenges to integrate air pollutant dynamics into a deep learning architecture? For example, is your contribution a new GNN with differential equations? How is it new? Or maybe is it a new method to handle two different simultaneous physical process?
> >
> > Maybe also helps: in contrast to differential equation-based traffic forecasting networks [1], what are the distinct technical challenges and insights?
> >
> > **Experiments:** thanks for the clarification. I think two cities are too few for properly validating a method like this. Note that current improvements over other methods are rather small. I'd suggest adding more datasets in the next resubmission.
> >
> > **Interpretability:** I just looked closer at the case study. The diffusion magnitude in Figure 4 could be done with any other GNN. In other words, you essentially use the same mechanism of a black box model (GNN) for interpretability. Therefore, interpretability is not something that your method enables, is it? If so, what is the added advantage of using the proposed method?
> >
> > [1] Jiahao Ji, Jingyuan Wang, Zhe Jiang, Jiawei Jiang, and Hu Zhang. Stden: Towards physics-guided neural networks for traffic flow prediction. In Proceedings of the AAAI Conference on Artificial Intelligence, volume 36, pp. 4048–4056, 2022.

---

> > > ### Author Response · Authors · 2023-11-22
> > > **Further Clarifications on Technical Challenges in comparison to the existing methods**
> > >
> > > We appreciate your follow-up comments and concerns regarding our work. Please find further clarifications below.
> > >
> > > **Specific Technical Challenges distinct to  Differential Equation-based Traffic Forecasting Networks:**
> > >
> > > While there are superficial similarities with differential equation-based traffic forecasting networks, the technical challenges and insights in our domain are distinct.
> > >
> > > 1. **Complexity of Physical Processes:**
> > >     * **Air Quality Modeling:** Involves understanding and modeling complex physical interactions in the atmosphere. This includes factors like the influence of meteorological conditions such as wind direction and wind speed considered in our study. These factors can have non-linear and interdependent, yet significant effects on air quality.
> > >     * **Traffic Forecasting:** While also complex, traffic forecasting predominantly deals with  more predictable patterns of vehicle movement influenced by factors like road capacity, traffic signals, and daily commuting patterns. These factors are generally more deterministic and less chaotic compared to atmospheric physics.
> > >
> > >     In [1], they have not considered any other external influencial factors, and only models the traffic flow based on the energy continuity equation assuming that the traffic flow is driven by latent potential energy fields. In contrast, our work uses the physics dynamics of air pollutant transport and models the pollutant concentration based on multiple interacting atmospheric processes encompassing factors such as meteorological conditions which adds a layer of complexity beyond the scope of traffic flow modeling.
> > >
> > >     Moreover, our model serves as a robust foundation for advancing air quality forecasts by facilitating the integration of additional external factors, including but not limited to chemical kinetics which we have mentioned as one of our future work.
> > >
> > > 2. **Data Variability and Uncertainty:**
> > >     * **Air Quality Data:** Often sparse, irregularly sampled, and subject to higher levels of uncertainty due to various sources of emissions and environmental influences. This makes it challenging to obtain a comprehensive and accurate dataset to train deep learning models.
> > >     * **Traffic Data:** Typically more regular and predictable. Data sources like GPS, traffic cameras, and sensors provide continuous and structured data, making it easier to model and predict traffic patterns.
> > >
> > > In our work, we have demonstrated the robust performance of our model across a spectrum of testing scenarios, including but not limited to sparse data prediction and the prediction of sudden changes (section 4.3). In contrast to traffic flow modeling [1], our model excels in handling the challenges posed by sparse data and abrupt changes, emphasizing its versatility in scenarios where traditional models may fall short.
> > >
> > > 3. **Spatial and Temporal Dynamics:**
> > >     * **Air Quality Modeling:** Requires modelling of complex spatiotemporal dynamics in pollutant concentrations as well as other influential factors (i.e wind data). In our study we use custom designed adjacency matrices and neural network layers to capture the spatiotemporal dynamics of wind and the respective effect on the concentration (section 3.3).
> > >     * **Traffic Flow Modelling**: While this also have spatial and temporal components, these are usually constrained by more strcutured road networks and deterministic human behavior patterns which are easier to model and predict.
> > >
> > >     In summary, the integration of air pollutant dynamics into a deep learning architecture is challenged mainly by the complexity of atmospheric processes, the variability and uncertainty in data, and intricate spatial-temporal dynamics. These challenges are distinct and often more complex compared to those encountered in traffic forecasting as explained above.
> > >
> > > [1] Jiahao Ji, Jingyuan Wang, Zhe Jiang, Jiawei Jiang, and Hu Zhang. Stden: Towards physics-guided neural networks for traffic flow prediction. In Proceedings of the AAAI Conference on Artificial Intelligence, volume 36, pp. 4048–4056, 2022.
> > >
> > > **Please proceed to the next window for the continuation of responses due to the character limitation per response.**

---

> ### Author Response · Authors · 2023-11-22
> **Technical Contributions in comparison to the existing methods, Clarification on Experiments and Interpretability**
>
> **Technical Contributions to integrate air pollutant dynamics into a deep learning architecture**
>
> As stated in our earlier response, the integration of physics priors within a deep learning architecture for air quality predictions, involves more than just introducing new equations or fusing exisitng architectures. In order to address the aforementioned distinct challenges, our contributions are as follows.
>
> * **Association of Physical Priors with Graph Geometry:** We have innovatively associated physical priors with the graph geometry to derive the diffusion-advection differential equations. This unique alignment allows for a more accurate representation of pollutant transport mechanisms across the sensor network. The derivation and integration of these equations within the graph structure are non-trivial, necessitating a deep understanding of both atmospheric physics and graph theory.
>
> * **Custom designed Adjacency Matrices and NN layers:** Central to our methodology is the development of custom NN layers and adjacency matrices that accurately model the derived diffusion-advection differential equation. These components were designed taking into account vital factors such as the capturing of spatiotemporal dependencies within the network and influence of wind patterns on air particle movement.
>
>     A critical aspect of this design is accommodating the influence of wind patterns on air particle movement. We introduced a novel Graph Neural Network (advGNN), specifically to model the advection process, drawing inspiration from molecular dynamics. This aspect of our model design is particularly innovative in its approach to representing complex environmental interactions.
>
> * **Fusion of multiple physical processes:** Recognizing the need to concurrently consider various physical processes, we undertook extensive experimentation with different fusion approaches. The implementation of a gated fusion mechanism emerged as the most effective strategy, demonstrating superior performance in integrating diverse elements of atmospheric dynamics.
>
> All in all, this is not a straightforward approach and we paid much attention and consideration to these critical factors. The successful integration of these components, within a physics guided deep learning architecture itself is a noteworthy advancement in addressing the intricate challenges posed by atmospheric processes. As to our knowledge, no existing methods have explored a similar fusion of physics priors with graph geometry, especially within the realm of air quality modeling.
>
> **Experiments:**
>
> We understand the importance of a diverse dataset for a thorough validation of our proposed method. As noted in our previous response, the primary reason for conducting experiments in a limited number of cities is the unavailability of comprehensive public datasets.We intend to explore collaborations and avenues to acquire additional datasets for future research.
>
> We acknowledge that the improvements over existing methods, may appear modest. However, it's important to note that our improvements are still significant and promising specially in diverse scenarios like sparse data and sudden change prediction in the field of environmental monitoring, where even small enhancements in predictive accuracy can have substantial practical implications for public health and policy-making.
>
> **Interpretability:**
>
> Your point about the diffusion magnitude being potentially replicable with other GNNs is well noted. Please find the clarification below on the unique aspects of our proposed method in terms of interpretability.
>
> * **Beyond Black Box Interpretability:** While it's true that the diffusion magnitude could be represented using standard GNN architectures, the key distinction of our model lies in its physics-guided framework. Unlike traditional GNNs, which often operate as 'black boxes', our model integrates specific physical laws and processes governing air pollutant dynamics. In other words, the GNN based network within our model is defined based on the diffusion and advection within the sensor network. This integration ensures that the model's learning and predictions are not just data-driven but also constrained and guided by established physical principles.
>
> * **Physics-Informed Feature Representations:** Our method enhances interpretability by generating physics-informed feature representations. This means that each node and edge in our network is not only a data point but also a representation of physical phenomena (like pollutant dispersion patterns). This approach provides a deeper understanding of how and why certain predictions are made, linking back to the underlying physical processes.

---

> ### Author Response · Authors · 2023-11-22
> **Request for Reconsideration of the score**
>
> Dear Reviewer pK4K,
>
> We sincerely appreciate the time and effort you have dedicated to reviewing our work. We have addressed the your questions and concerns above. Given that the end of author/reviewer discussions is just in few hours, this is to kindly ask for your reconsideration of the score. We would really appreciate it if our next round of communication could leave time for us to resolve any of your remaining or new questions.
>
> Thank you so much for devoting time to improving our work!

---

> ### Comment · Reviewer_pK4K · 2023-11-22
> **Response**
>
> Thanks for the thorough response. I haven't gone through it in detail yet, and will do so later. Then I will definitely reconsider my score.
>
> If I may ask one last clarification question: Can you please point me to where in the case study has been validated your claim "our method enhances interpretability by physics-informed feature representations"?
>
> I agree with the general intuition that physics-informed representations will help with more interpretable predictions. However, I do not see this potential being used in your experiments and/or case study. Maybe I missed it. Thanks!

---

> > ### Author Response · Authors · 2023-11-22
> > **Clarification on Interpretability**
> >
> > Thank you for your follow-up question and the opportunity to provide further clarification physics-informed feature representations enhancing interpretability. While the explicit phrase "our method enhances interpretability by physics-informed feature representations" may not be explicitly stated in the text, the following sections illustrate the practical application of this concept:
> >
> > * **Diffusion Advection Differential Equation:** In the section 3.3, which defines the two physical processes on the sensor network, we state how the nodes and edges within the network are representations of physical phenomena (like pollutant dispersion patterns). Furthermore, we highlight the incorporation of physical laws governing air pollutant dynamics into the GNN structure within this section as well as in section 3.4.
> >
> >     By defining the network based on diffusion and advection processes within the sensor network, we ensure that the model's representations are not purely data-driven but also grounded in the physics of pollutant dispersion.
> >
> > * **Case Study**: When discussing specific predictions or cases in the case study, we provide insights into how the physics-informed features contribute to the interpretability of the model's output. Specifically, we emphasize the significance of understanding the physical processes represented by each node and edge in the network, linking the predictions back to the underlying physics.
> >
> > While we acknowledge that we might not have used the exact wording explicitly, we are confident that the reasoning based on inclusion of physics-guided principles in the case study supports our assertion of improved interpretability of the model.
> >
> > Hope this clarifies on your concern further. Thank you once again for your valuable feedback towards enhancing our work!

---

> ### Author Response · Authors · 2023-11-23
> **Reminder for Re-evaluation and Score Raising**
>
> Dear Reviewer pK4K,
>
> This is to kindly remind you to reconsider our score as we are approaching the end of the rebuttal period in a few hours. We have provided clarification on your concerns above.  We sincerely appreciate the time and effort you have dedicated to reviewing our work.
>
> Thanks again for your valuable suggestions and comments for improving our work !
>
> Best Regards,
> Authors

---

### Author Response · Authors · 2023-11-21
**Rebuttal and Revised Paper Submission**

Dear Reviewers,

Thanks again for your detailed review and constructive suggestions for improvement. We have carefully considered your comments and suggestions and provided detailed responses below. A new version of our paper has been uploaded with modified parts colored red, based on the rebuttal.

As the rebuttal period is concluding tomorrow, we kindly request you to revisit our responses at your earliest convenience. We are hopeful that our responses adequately address your concerns and provide clarity on the modifications made to the manuscript.

If you require further clarification on any points, please do not hesitate to reach out. We are willing to provide additional information to ensure your confidence in the revisions made.

Sincerely, The Authors

---

### Author Response · Authors · 2023-11-22
**General Response to all Reviewers and Major Revisions**

We sincerely thank all the reviewers for their detailed review and constructive suggestions for improvement. We appreciate that almost all reviewers recognized the novelty, significance of our work and the robust performance of our model across diverse testing scenarios. We have carefully considered all the comments and suggestions and provided detailed responses below. A new version of our paper has been uploaded with modified parts colored red, based on the rebuttal. Here we highlight our major revisions:

* In response to questions and feedback from **Reviewer pK4K**, we highlighted the unique challenges in the air quality prediction problem and elaborated on the technical contributions of our work explicitly noting the significance of the work for ICLR's research track.Furthermore, we included further clarifications on concerns regarding experimentation and interpretability.

* In response to **Reviewer 6p5Q**'s feedback, we further clarified on the research question and literature review and moved the Related Work section after the Introduction and before the Methodology in our revised manuscript. We further enhanced the related work section with more recent citations.Furthermore, we included a Discussion section in the revised manuscript in the Appendix (A.7). The discussion section includes summary of results, implications of the findings in general, limitations and few recommendations for further enhancement.We also highlighted these limitations within section 5 (Conclusion and Future Work)

* Following feedback from **Reviewer 8KYF**, we provided clarifications on physical processes, the reparameterization trick and included detailed advantages of integrating physical principles into deep learning Models. Moreover, the typos and minor spacing issues have been resolved in the revised manuscript.

* In response to feedback from **Reviewer tKfZ**, we addressed all the experimental concerns comprehensively, provided comparison with CTM and clarification of effect of model components. We further provided some insights to model scalability by conducting experiments on a much larger dataset. Furthermore we addressed reviewer concerns related to the dataset, providing insights though an exploratory data analysis.

---

### Meta-Review · Area_Chair_HbHQ · 2023-12-05

**Metareview:**

The manuscript has been reviewed by 4 experts with reviews converging towards a positive consensus on the paper's acceptance. The reviewers appreciate the innovative approach of integrating physical principles into a neural network for air quality prediction, recognizing its potential to contribute significantly to both the machine learning and environmental science communities. I agree with the reviewers and recommend acceptance. I strongly urge authors to incorporate reviewer comments. Specifically,


Clarify Technical Contributions: I recommend that the authors to clarify the novel technical contributions made in this work, especially those discussed during the paper discussion phase.

Expand Experiments: Consider extending the experimental setup to more cities, if feasible, to enhance the robustness and generalizability of the results.

Enhance Interpretability Discussion: Provide a deeper analysis of the model's interpretability. Demonstrate how the physical principles integrated into the model are leveraged for predictions.

**Justification For Why Not Higher Score:**

The paper is definitely interesting and worth to be published. However, it has no major algorithmic contribution. Hence, it is only impactful to the subset of the community.

**Justification For Why Not Lower Score:**

All the reviewers and AC agree on the correctness and quality of the manuscript. Hence, it should be shared with the community.

---

### Decision · Program_Chairs · 2024-01-16

Accept (poster)